# Optimal Cluster Recovery
# in the Labeled Stochastic Block Model

**Se-Young Yun**
CNLS, Los Alamos National Lab.
Los Alamos, NM 87545
syun@lanl.gov

**Alexandre Proutiere**
Automatic Control Dept., KTH
Stockholm 100-44, Sweden
alepro@kth.se

## Abstract

We consider the problem of community detection or clustering in the labeled Stochastic Block Model (LSBM) with a finite number $K$ of clusters of sizes linearly growing with the global population of items $n$. Every pair of items is labeled independently at random, and label $\ell$ appears with probability $p(i, j, \ell)$ between two items in clusters indexed by $i$ and $j$, respectively. The objective is to reconstruct the clusters from the observation of these random labels.

Clustering under the SBM and their extensions has attracted much attention recently. Most existing work aimed at characterizing the set of parameters such that it is possible to infer clusters either positively correlated with the true clusters, or with a vanishing proportion of misclassified items, or exactly matching the true clusters. We find the set of parameters such that there exists a clustering algorithm with at most $s$ misclassified items in average under the general LSBM and for any $s = o(n)$, which solves one open problem raised in [2]. We further develop an algorithm, based on simple spectral methods, that achieves this fundamental performance limit within $O(n\text{polylog}(n))$ computations and without the a-priori knowledge of the model parameters.

## 1 Introduction

Community detection consists in extracting (a few) groups of similar items from a large global population, and has applications in a wide spectrum of disciplines including social sciences, biology, computer science, and statistical physics. The communities or clusters of items are inferred from the observed pair-wise similarities between items, which, most often, are represented by a graph whose vertices are items and edges are pairs of items known to share similar features.

The stochastic block model (SBM), introduced three decades ago in [12], constitutes a natural performance benchmark for community detection, and has been, since then, widely studied. In the SBM, the set of items $\mathcal{V} = \{1, \ldots, n\}$ are partitioned into $K$ non-overlapping clusters $\mathcal{V}_1, \ldots, \mathcal{V}_K$, that have to be recovered from an observed realization of a random graph. In the latter, an edge between two items belonging to clusters $\mathcal{V}_i$ and $\mathcal{V}_j$, respectively, is present with probability $p(i, j)$, independently of other edges. The analyses presented in this paper apply to the SBM, but also to the *labeled* stochastic block model (LSBM) [11], a more general model to describe the similarities of items. There, the observation of the similarity between two items comes in the form of a *label* taken from a finite set $\mathcal{L} = \{0, 1, \ldots, L\}$, and label $\ell$ is observed between two items in clusters $\mathcal{V}_i$ and $\mathcal{V}_j$, respectively, with probability $p(i, j, \ell)$, independently of other labels. The standard SBM can be seen as a particular instance of its labeled counterpart with two possible labels 0 and 1, and where the edges present (resp. absent) in the SBM correspond to item pairs with label 1 (resp. 0). The problem of cluster recovery under the LSBM consists in inferring the hidden partition $\mathcal{V}_1, \ldots, \mathcal{V}_K$ from the observation of the random labels on each pair of items.

Over the last few years, we have seen remarkable progresses for the problem of cluster recovery under the SBM (see [7] for an exhaustive literature review), highlighting its scientific relevance and richness. Most recent work on the SBM aimed at characterizing the set of parameters (i.e., the probabilities $p(i, j)$ that there exists an edge between nodes in clusters $i$ and $j$ for $1 \le i, j \le K$) such that some qualitative recovery objectives can or cannot be met. For sparse scenarios where the average degree of items in the graph is $O(1)$, parameters under which it is possible to extract clusters positively correlated with the true clusters have been identified [5, 18, 16]. When the average degree of the graph is $\omega(1)$, one may predict the set of parameters allowing a cluster recovery with a vanishing (as $n$ grows large) proportion of misclassified items [22, 17], but one may also characterize parameters for which an asymptotically exact cluster reconstruction can be achieved [1, 21, 8, 17, 2, 3, 13].

In this paper, we address the finer and more challenging question of determining, under the general LSBM, the minimal number of misclassified items given the parameters of the model. Specifically, for any given $s = o(n)$, our goal is to identify the set of parameters such that it is possible to devise a clustering algorithm with at most $s$ misclassified items. Of course, if we achieve this goal, we shall recover all the aforementioned results on the SBM.

**Main results.** We focus on the labeled SBM as described above, and where each item is assigned to cluster $\mathcal{V}_k$ with probability $\alpha_k > 0$, independently of other items. We assume w.l.o.g. that $\alpha_1 \le \alpha_2 \le \cdots \le \alpha_K$. We further assume that $\alpha = (\alpha_1, \ldots, \alpha_K)$ does not depend on the total population of items $n$. Conditionally on the assignment of items to clusters, the pair or edge $(v, w) \in \mathcal{V}^2$ has label $\ell \in \mathcal{L} = \{0, 1, \ldots, L\}$ with probability $p(i, j, \ell)$, when $v \in \mathcal{V}_i$ and $w \in \mathcal{V}_j$. W.l.o.g., 0 is the most frequent label, i.e., $0 = \arg\max_\ell \sum_{i=1}^{K} \sum_{j=1}^{K} \alpha_i \alpha_j p(i, j, \ell)$. Throughout the paper, we typically assume that $\bar{p} = o(1)$ and $\bar{p}n = \omega(1)$ where $\bar{p} = \max_{i, j, \ell \ge 1} p(i, j, \ell)$ denotes the maximum probability of observing a label different than 0. We shall explicitly state whether these assumption are made when deriving our results. In the standard SBM, the second assumption means that the average degree of the corresponding random graph is $\omega(1)$. This also means that we can hope to recover clusters with a vanishing proportion of misclassified items. We finally make the following assumption: there exist positive constants $\eta$ and $\varepsilon$ such that for every $i, j, k \in [K] = \{1, \ldots, K\}$,

$$\text{(A1)} \quad \forall \ell \in \mathcal{L}, \ \frac{p(i, j, \ell)}{p(i, k, \ell)} \le \eta \quad \text{and} \quad \text{(A2)} \quad \frac{\sum_{k=1}^{K} \sum_{\ell=1}^{L} (p(i, k, \ell) - p(j, k, \ell))^2}{\bar{p}^2} \ge \varepsilon.$$

(A2) imposes a certain separation between the clusters. For example, in the standard SBM with two communities, $p(1, 1, 1) = p(2, 2, 1) = \xi$, and $p(1, 2, 1) = \zeta$, (A2) is equivalent to $2(\xi - \zeta)^2 / \xi^2 \ge \epsilon$. In summary, the LSBM is parametrized by $\alpha$ and $p = (p(i, j, \ell))_{1 \le i, j \le K, 0 \le \ell \le L}$, and recall that $\alpha$ does not depend on $n$, whereas $p$ does.

For the above LSBM, we derive, for any arbitrary $s = o(n)$, a necessary condition under which there exists an algorithm inferring clusters with $s$ misclassified items. We further establish that under this condition, a simple extension of spectral algorithms extract communities with less than $s$ misclassified items. To formalize these results, we introduce the *divergence* of $(\alpha, p)$. We denote by $p(i)$ the $K \times (L + 1)$ matrix whose element on the $j$-th row and the $(\ell + 1)$-th column is $p(i, j, \ell)$-th and denote by $p(i, j) \in [0, 1]^{L+1}$ the vector describing the probability distribution of the label of a pair of items in $\mathcal{V}_i$ and $\mathcal{V}_j$, respectively. Let $\mathcal{P}^{K \times (L+1)}$ denote the set of $K \times (L + 1)$ matrices such that each row represents a probability distribution. The divergence $D(\alpha, p)$ of $(\alpha, p)$ is defined as follows: $D(\alpha, p) = \min_{i, j: i \ne j} D_{L+}(\alpha, p(i), p(j))$ with

$$D_{L+}(\alpha, p(i), p(j)) = \min_{y \in \mathcal{P}^{K \times (L+1)}} \max \left\{ \sum_{k=1}^{K} \alpha_k KL(y(k), p(i, k)), \sum_{k=1}^{K} \alpha_k KL(y(k), p(j, k)) \right\}$$

where $KL$ denotes the Kullback-Leibler divergence between two label distributions, i.e., $KL(y(k), p(i, k)) = \sum_{\ell=0}^{L} y(k, \ell) \log \frac{y(k, \ell)}{p(i, k, \ell)}$. Finally, we denote by $\varepsilon^\pi(n)$ the number of misclassified items under the clustering algorithm $\pi$, and by $\mathbb{E}[\varepsilon^\pi(n)]$ its expectation (with respect to the randomness in the LSBM and in the algorithm).

We first derive a tight lower bound on the average number of misclassified items when the latter is $o(n)$. Note that such a bound was unknown even for the SBM [2].

**Theorem 1** *Assume that (A1) and (A2) hold, and that $\bar{p}n = \omega(1)$. Let $s = o(n)$. If there exists a clustering algorithm $\pi$ misclassifying in average less than $s$ items asymptotically, i.e.,*

$\limsup_{n\to\infty} \frac{\mathbb{E}[\varepsilon^{\pi}(n)]}{s} \leq 1$, *then the parameters* $(\alpha, p)$ *of the LSBM satisfy:*

$$\lim_{n\to\infty} \inf \frac{nD(\alpha, p)}{\log(n/s)} \geq 1. \tag{1}$$

To state the corresponding positive result (i.e., the existence of an algorithm misclassifying only $s$ items), we make an additional assumption to avoid extremely sparse labels: (A3) there exists a constant $\kappa > 0$ such that $np(j, i, \ell) \geq (n\bar{p})^{\kappa}$ for all $i, j$ and $\ell \geq 1$.

**Theorem 2** *Assume that (A1), (A2), and (A3) hold, and that* $\bar{p} = o(1)$, $\bar{p}n = \omega(1)$. *Let* $s = o(n)$. *If the parameters* $(\alpha, p)$ *of the LSBM satisfy (1), then the Spectral Partition (SP) algorithm presented in Section 4 misclassifies at most* $s$ *items with high probability, i.e.,* $\lim_{n\to\infty} \mathbb{P}[\varepsilon^{SP}(n) \leq s] = 1$.

These theorems indicate that under the LSBM with parameters satisfying (A1) and (A2), the number of misclassified items scales at least as $n \exp(-nD(\alpha, p)(1 + o(1)))$ under any clustering algorithm, irrespective of its complexity. They further establish that the Spectral Partition algorithm reaches this fundamental performance limit under the additional condition (A3). We note that the SP algorithm runs in polynomial time, i.e., it requires $O(n^2\bar{p}\log(n))$ floating-point operations.

We further establish a necessary and sufficient condition on the parameters of the LSBM for the existence of a clustering algorithm recovering the clusters exactly with high probability. Deriving such a condition was also open [2].

**Theorem 3** *Assume that (A1) and (A2) hold. If there exists a clustering algorithm that does not misclassify any item with high probability, then the parameters* $(\alpha, p)$ *of the LSBM satisfy:* $\liminf_{n\to\infty} \frac{nD(\alpha, p)}{\log(n)} \geq 1$. *If this condition holds, then under (A3), the SP algorithm recovers the clusters exactly with high probability.*

The paper is organized as follows. Section 2 presents the related work and example of application of our results. In Section 3, we sketch the proof of Theorem 1, which leverages change-of-measure and coupling arguments. We present in Section 4 the Spectral Partition algorithm, and analyze its performance (we outline the proof of Theorem 2). All results are proved in details in the supplementary material.

## 2 Related Work and Applications

### 2.1 Related work

Cluster recovery in the SBM has attracted a lot of attention recently. We summarize below existing results, and compare them to ours. Results are categorized depending on the targeted level of performance. First, we consider the notion of *detectability*, the lowest level of performance requiring that the extracted clusters are just positively correlated with the true clusters. Second, we look at *asymptotically accurate recovery*, stating that the proportion of misclassified items vanishes as $n$ grows large. Third, we present existing results regarding exact cluster recovery, which means that no item is misclassified. Finally, we report recent work whose objective, like ours, is to characterize the optimal cluster recovery rate.

**Detectability.** Necessary and sufficient conditions for *detectability* have been studied for the binary symmetric SBM (i.e., $L = 1$, $K = 2$, $\alpha_1 = \alpha_2$, $p(1, 1, 1) = p(2, 2, 1) = \xi$, and $p(1, 2, 1) = p(2, 1, 1) = \zeta$). In the sparse regime where $\xi, \zeta = o(1)$, and for the binary symmetric SBM, the main focus has been on identifying the phase transition threshold (a condition on $\xi$ and $\zeta$) for *detectability*: It was conjectured in [5] that if $n(\xi - \zeta) < \sqrt{2n(\xi + \zeta)}$ (i.e., under the threshold), no algorithm can perform better than a simple random assignment of items to clusters, and above the threshold, clusters can partially be recovered. The conjecture was recently proved in [18] (necessary condition), and [16] (sufficient condition). The problem of detectability has been also recently studied in [24] for the asymmetric SBM with more than two clusters of possibly different sizes. Interestingly, it is shown that in most cases, the phase transition for detectability disappears.

The present paper is not concerned with conditions for detectability. Indeed detectability means that only a strictly positive proportion of items can be correctly classified, whereas here, we impose that the proportion of misclassified items vanishes as $n$ grows large.

**Asymptotically accurate recovery.** A necessary and sufficient condition for asymptotically accurate recovery in the SBM (with any number of clusters of different but linearly increasing sizes) has been derived in [22] and [17]. Using our notion of divergence specialized to the SBM, this condition is $nD(\alpha, p) = \omega(1)$. Our results are more precise since the minimal achievable number of misclassified items is characterized, and apply to a broader setting since they are valid for the generic LSBM.

**Asymptotically exact recovery.** Conditions for exact cluster recovery in the SBM have been also recently studied. [1, 17, 8] provide a necessary and sufficient condition for asymptotically exact recovery in the binary symmetric SBM. For example, it is shown that when $\xi = \frac{a \log(n)}{n}$ and $\zeta = \frac{b \log(n)}{n}$ for $a > b$, clusters can be recovered exactly if and only if $\frac{a+b}{2} - \sqrt{ab} \geq 1$. In [2, 3], the authors consider a more general SBM corresponding to our LSBM with $L = 1$. They define CH-divergence as:

$$D_+(\alpha, p(i), p(j))$$
$$= \frac{n}{\log(n)} \max_{\lambda \in [0,1]} \sum_{k=1}^{K} \alpha_k \left( (1-\lambda)p(i,k,1) + \lambda p(j,k,1) - p(i,k,1)^{1-\lambda} p(j,k,1)^{\lambda} \right),$$

and show that $\min_{i \neq j} D_+(\alpha, p(i), p(j)) > 1$ is a necessary and sufficient condition for asymptotically exact reconstruction. The following claim, proven in the supplementary material, relates $D_+$ to $D_{L+}$.

**Claim 4** *When $\bar{p} = o(1)$, we have for all $i, j$:*

$$D_{L+}(\alpha, p(i), p(j))$$
$$\stackrel{n \to \infty}{\sim} \max_{\lambda \in [0,1]} \sum_{\ell=1}^{L} \sum_{k=1}^{K} \alpha_k \left( (1-\lambda)p(i,k,\ell) + \lambda p(j,k,\ell) - p(i,k,\ell)^{1-\lambda} p(j,k,\ell)^{\lambda} \right).$$

Thus, the results in [2, 3] are obtained by applying Theorem 3 and Claim 4.

In [13], the authors consider a symmetric labeled SBM where communities are balanced (i.e., $\alpha_k = \frac{1}{K}$ for all $k$) and where label probabilities are simply defined as $p(i,i,\ell) = p(\ell)$ for all $i$ and $p(i,j,\ell) = q(\ell)$ for all $i \neq j$. It is shown that $\frac{nI}{\log(n)} > 1$ is necessary and sufficient for asymptotically exact recovery, where $I = -\frac{2}{K} \log \left( \sum_{\ell=0}^{L} \sqrt{p(\ell)q(\ell)} \right)$. We can relate $I$ to $D(\alpha, p)$:

**Claim 5** *In the LSBM with $K$ clusters, if $\bar{p} = o(1)$, and for all $i, j, \ell$ such that $i \neq j$, $\alpha_i = \frac{1}{K}$, $p(i,i,\ell) = p(\ell)$, and $p(j,k,\ell) = q(\ell)$, we have:* $D(\alpha, p) \stackrel{n \to \infty}{\sim} -\frac{2}{K} \log \left( \sum_{\ell=0}^{L} \sqrt{p(\ell)q(\ell)} \right).$

Again from this claim, the results derived in [13] are obtained by applying Theorem 3 and Claim 5.

**Optimal recovery rate.** In [6, 19], the authors consider the binary SBM in the sparse regime where the average degree of items in the graph is $O(1)$, and identify the minimal number of misclassified items for very specific intra- and inter-cluster edge probabilities $\xi$ and $\zeta$. Again the sparse regime is out of the scope of the present paper. [23, 7] are concerned with the general SBM corresponding to our LSBM with $L = 1$, and with regimes where asymptotically accurate recovery is possible. The authors first characterize the optimal recovery rate in a minimax framework. More precisely, they consider a (potentially large) set of possible parameters $(\alpha, p)$, and provide a lower bound on the expected number of misclassified items for the worst parameters in this set. Our lower bound (Theorem 1) is more precise as it is model-specific, i.e., we provide the minimal expected number of misclassified items for a given parameter $(\alpha, p)$ (and for a more general class of models). Then the authors propose a clustering algorithm, with time complexity $O(n^3 \log(n))$, and achieving their minimax recovery rate. In comparison, our algorithm yields an optimal recovery rate $O(n^2 \bar{p} \log(n))$ for any given parameter $(\alpha, p)$, exhibits a lower running time, and applies to the generic LSBM.

## 2.2 Applications

We provide here a few examples of application of our results, illustrating their versatility. In all examples, $f(n)$ is a function such that $f(n) = \omega(1)$, and $a, b$ are fixed real numbers such that $a > b$.

**The binary SBM.** Consider the binary SBM where the average item degree is $\Theta(f(n))$, and represented by a LSBM with parameters $L = 1, K = 2, \alpha = (\alpha_1, 1-\alpha_1), p(1,1,1) = p(2,2,1) = \frac{af(n)}{n}$, and $p(1,2,1) = p(2,1,1) = \frac{bf(n)}{n}$. From Theorems 1 and 2, the optimal number of misclassified vertices scales as $n \exp(-g(\alpha_1, a, b)f(n)(1 + o(1)))$ when $\alpha_1 \leq 1/2$ (w.l.o.g.) and where

$$g(\alpha_1, a, b) := \max_{\lambda \in [0,1]} (1 - \alpha_1 - \lambda + 2\alpha_1\lambda)a + (\alpha_1 + \lambda - 2\alpha\lambda)b - \alpha_1 a^\lambda b^{(1-\lambda)} - (1 - \alpha_1)a^{(1-\lambda)}b^\lambda.$$

It can be easily checked that $g(\alpha_1, a, b) \geq g(1/2, a, b) = \frac{1}{2}(\sqrt{a} - \sqrt{b})^2$ (letting $\lambda = \frac{1}{2}$). The worst case is hence obtained when the two clusters are of equal sizes. When $f(n) = \log(n)$, we also note that the condition for asymptotically exact recovery is $g(\alpha_1, a, b) \geq 1$.

**Recovering a single hidden community.** As in [9], consider a random graph model with a hidden community consisting of $\alpha n$ vertices, edges between vertices belonging the hidden community are present with probability $\frac{af(n)}{n}$, and edges between other pairs are present with probability $\frac{bf(n)}{n}$. This is modeled by a LSBM with parameters $K = 2, L = 1, \alpha_1 = \alpha, p(1,1,1) = \frac{af(n)}{n}$, and $p(1,2,1) = p(2,1,1) = p(2,2,1) = \frac{bf(n)}{n}$. The minimal number of misclassified items when searching for the hidden community scales as $n \exp(-h(\alpha, a, b)f(n)(1 + o(1)))$ where

$$h(\alpha, a, b) := \alpha \left( a - (a - b)\frac{1 + \log(a - b) - \log(a\log(a/b))}{\log(a/b)} \right).$$

When $f(n) = \log(n)$, the condition for asymptotically exact recovery of the hidden community is $h(\alpha, a, b) \geq 1$.

**Optimal sampling for community detection under the SBM.** Consider a dense binary symmetric SBM with intra- and inter-cluster edge probabilities $a$ and $b$. In practice, to recover the clusters, one might not be able to observe the entire random graph, but sample its vertex (here item) pairs as considered in [22]. Assume for instance that any pair of vertices is sampled with probability $\frac{\delta f(n)}{n}$ for some fixed $\delta > 0$, independently of other pairs. We can model such scenario using a LSBM with three labels, namely $\times$, 0 and 1, corresponding to the absence of observation (the vertex pair is not sampled), the observation of the absence of an edge and of the presence of an edge, respectively, and with parameters for all $i, j \in \{1, 2\}, p(i, j, \times) = 1 - \frac{\delta f(n)}{n}, p(1,1,1) = p(2,2,1) = a\frac{\delta f(n)}{n}$, and $p(1,2,1) = p(2,1,1) = b\frac{\delta f(n)}{n}$. The minimal number of misclassified vertices scales as $n \exp(-l(\delta, a, b)f(n)(1 + o(1)))$ where $l := \delta(1 - \sqrt{ab} - \sqrt{(1-a)(1-b)})$. When $f(n) = \log(n)$, the condition for asymptotically exact recovery is $l(\alpha, a_+, a_-, b_+, b_-) \geq 1$.

**Signed networks.** Signed networks [15, 20] are used in social sciences to model positive and negative interactions between individuals. These networks can be represented by a LSBM with three possible labels, namely 0, + and -, corresponding to the absence of interaction, positive and negative interaction, respectively. Consider such LSBM with parameters: $K = 2, \alpha_1 = \alpha_2, p(1,1,+) = p(2,2,+) = \frac{a_+f(n)}{n}, p(1,1,-) = p(2,2,-) = \frac{a_-f(n)}{n}, p(1,2,+) = p(2,1,+) = \frac{b_+f(n)}{n}$, and $p(1,2,-) = p(2,1,-) = \frac{b_-f(n)}{n}$, for some fixed $a_+, a_-, b_+, b_-$ such that $a_+ > b_+$ and $a_- < b_-$. The minimal number of misclassified individuals here scales as $n \exp(-m(\alpha, a_+, a_-, b_+, b_-)f(n)(1 + o(1)))$ where

$$m(\alpha, a_+, a_-, b_+, b_-) := \frac{1}{2} \left( (\sqrt{a_+} - \sqrt{b_+})^2 + (\sqrt{a_-} - \sqrt{b_-})^2 \right).$$

When $f(n) = \log(n)$, the condition for asymptotically exact recovery is $l(\alpha, a_+, a_-, b_+, b_-) \geq 1$.

## 3 Fundamental Limits: Change of Measures through Coupling

In this section, we explain the construction of the proof of Theorem 1. The latter relies on an appropriate *change-of-measure* argument, frequently used to identify upper performance bounds in

online stochastic optimization problems [14]. In the following, we refer to $\Phi$, defined by parameters $(\alpha, p)$, as the true stochastic model under which all the observed random labels are generated, and denote by $\mathbb{P}_\Phi = \mathbb{P}$ (resp. $\mathbb{E}_\Phi[\cdot] = \mathbb{E}[\cdot]$) the corresponding probability measure (resp. expectation). In our change-of-measure argument, we construct a second stochastic model $\Psi$ (whose corresponding probability measure and expectation are $\mathbb{P}_\Psi$ and $\mathbb{E}_\Psi[\cdot]$, respectively). Using a change of measures from $\mathbb{P}_\Phi$ to $\mathbb{P}_\Psi$, we relate the expected number of misclassified items $\mathbb{E}_\Phi[\varepsilon^\pi(n)]$ under any clustering algorithm $\pi$ to the expected (w.r.t. $\mathbb{P}_\Psi$) log-likelihood ratio $\mathcal{Q}$ of the observed labels under $\mathbb{P}_\Phi$ and $\mathbb{P}_\Psi$. Specifically, we show that, roughly, $\log(n/\mathbb{E}_\Phi[\varepsilon^\pi(n)])$ must be smaller than $\mathbb{E}_\Psi[\mathcal{Q}]$ for $n$ large enough.

**Construction of** $\psi$. Let $(i^\star, j^\star) = \arg\min_{i,j:i<j} D_{L+}(\alpha, p(i), p(j))$, and let $v^\star$ denote the smallest item index that belongs to cluster $i^\star$ or $j^\star$. If both $\mathcal{V}_{i^\star}$ and $\mathcal{V}_{j^\star}$ are empty, we define $v^\star = n$. Let $q \in \mathcal{P}^{K \times (L+1)}$ such that: $D(\alpha, p) = \sum_{k=1}^K \alpha_k KL(q(k), p(i^\star, k)) = \sum_{k=1}^K \alpha_k KL(q(k), p(j^\star, k))$. The existence of such $q$ is proved in Lemma 7 in the supplementary material. Now to define the stochastic model $\Psi$, we couple the generation of labels under $\Phi$ and $\Psi$ as follows.

1. We first generate the random clusters $\mathcal{V}_1, \ldots, \mathcal{V}_K$ under $\Phi$, and extract $i^\star$, $j^\star$, and $v^\star$. The clusters generated under $\Psi$ are the same as those generated under $\Phi$. For any $v \in \mathcal{V}$, we denote by $\sigma(v)$ the cluster of item $v$.

2. For all pairs $(v, w)$ such that $v \neq v^\star$ and $w \neq v^\star$, the labels generated under $\Psi$ are the same as those generated under $\Phi$, i.e., the label $\ell$ is observed on the edge $(v, w)$ with probability $p(\sigma(v), \sigma(w), \ell)$.

3. Under $\Psi$, for any $v \neq v^\star$, the observed label on the edge $(v, v^\star)$ under $\Psi$ is $\ell$ with probability $q(\sigma(v), \ell)$.

Let $x_{v,w}$ denote the label observed for the pair $(v, w)$. We introduce $\mathcal{Q}$, the log-likelihood ratio of the observed labels under $\mathbb{P}_\Phi$ and $\mathbb{P}_\Psi$ as:

$$\mathcal{Q} = \sum_{v=1}^{v^\star-1} \log \frac{q(\sigma(v), x_{v^\star,v})}{p(\sigma(v^\star), \sigma(v), x_{v^\star,v})} + \sum_{v=v^\star+1}^{n} \log \frac{q(\sigma(v), x_{v^\star,v})}{p(\sigma(v^\star), \sigma(v), x_{v^\star,v})}. \tag{2}$$

Let $\pi$ be a clustering algorithm with output $(\hat{\mathcal{V}}_k)_{1 \leq k \leq K}$, and let $\mathcal{E} = \bigcup_{1 \leq k \leq K} \hat{\mathcal{V}}_k \setminus \mathcal{V}_k$ be the set of misclassified items under $\pi$. Note that in general in our analysis, we always assume without loss of generality that $|\bigcup_{1 \leq k \leq K} \hat{\mathcal{V}}_k \setminus \mathcal{V}_k| \leq |\bigcup_{1 \leq k \leq K} \hat{\mathcal{V}}_{\gamma(k)} \setminus \mathcal{V}_k|$ for any permutation $\gamma$, so that the set of misclassified items is indeed $\mathcal{E}$. By definition, $\varepsilon^\pi(n) = |\mathcal{E}|$. Since under $\Phi$, items are interchangeable (remember that items are assigned to the various clusters in an i.i.d. manner), we have: $n\mathbb{P}_\Phi\{v \in \mathcal{E}\} = \mathbb{E}_\Phi[\varepsilon^\pi(n)] = \mathbb{E}[\varepsilon^\pi(n)]$.

Next, we establish a relationship between $\mathbb{E}[\varepsilon^\pi(n)]$ and the distribution of $\mathcal{Q}$ under $\mathbb{P}_\Psi$. For any function $f(n)$, we can prove that: $\mathbb{P}_\Psi\{\mathcal{Q} \leq f(n)\} \leq \exp(f(n))\frac{\mathbb{E}_\Phi[\varepsilon^\pi(n)]}{(\alpha_{i^\star}+\alpha_{j^\star})n} + \frac{\alpha_{j^\star}}{\alpha_{i^\star}+\alpha_{j^\star}}$. Using this result with $f(n) = \log(n/\mathbb{E}_\Phi[\varepsilon^\pi(n)]) - \log(2/\alpha_{i^\star})$, and Chebyshev's inequality, we deduce that: $\log(n/\mathbb{E}_\Phi[\varepsilon^\pi(n)]) - \log(2/\alpha_{i^\star}) \leq \mathbb{E}_\Psi[\mathcal{Q}] + \sqrt{\frac{4}{\alpha_{i^\star}}\mathbb{E}_\Psi[(\mathcal{Q} - \mathbb{E}_\Psi[\mathcal{Q}])^2]}$, and thus, a necessary condition for $\mathbb{E}[\varepsilon^\pi(n)] \leq s$ is:

$$\log(n/s) - \log(2/\alpha_{i^\star}) \leq \mathbb{E}_\Psi[\mathcal{Q}] + \sqrt{\frac{4}{\alpha_{i^\star}}\mathbb{E}_\Psi[(\mathcal{Q} - \mathbb{E}_\Psi[\mathcal{Q}])^2]}. \tag{3}$$

**Analysis of** $\mathcal{Q}$. In view of (3), we can obtain a necessary condition for $\mathbb{E}[\varepsilon^\pi(n)] \leq s$ if we evaluate $\mathbb{E}_\Psi[\mathcal{Q}]$ and $\mathbb{E}_\Psi[(\mathcal{Q} - \mathbb{E}_\Psi[\mathcal{Q}])^2]$. To evaluate $\mathbb{E}_\Psi[\mathcal{Q}]$, we can first prove that $v^\star \leq \log(n)^2$ with high probability. From this, we can approximate $\mathbb{E}_\Psi[\mathcal{Q}]$ by $\mathbb{E}_\Psi[\sum_{v=v^\star+1}^{n} \log \frac{q(\sigma(v), x_{v^\star,v})}{p(\sigma(v^\star), \sigma(v), x_{v^\star,v})}]$, which is itself well-approximated by $nD(\alpha, p)$. More formally, we can show that:

$$\mathbb{E}_\Psi[\mathcal{Q}] \leq (n + 2\log(\eta)\log(n)^2) D(\alpha, p) + \frac{\log \eta}{n^3}. \tag{4}$$

Similarly, we prove that $\mathbb{E}_\Psi[(\mathcal{Q} - \mathbb{E}_\Psi[\mathcal{Q}])^2] = O(n\bar{p})$, which in view of Lemma 8 (refer to the supplementary material) and assumption (A2), implies that: $\mathbb{E}_\Psi[(\mathcal{Q} - \mathbb{E}_\Psi[\mathcal{Q}])^2] = o(nD(\alpha, p))$.

We complete the proof of Theorem 1 by putting the above arguments together: From (3), (4) and the above analysis of $\mathcal{Q}$, when the expected number of misclassified items is less than $s$ (i.e., $\mathbb{E}[\varepsilon^\pi(n)] \leq s$), we must have: $\liminf_{n\to\infty} \frac{nD(\alpha,p)}{\log(n/s)} \geq 1$.

## 4 The Spectral Partition Algorithm and its Optimality

In this section, we sketch the proof of Theorem 2. To this aim, we present the Spectral Partition (SP) algorithm and analyze its performance. The SP algorithm consists in two parts, and its detailed pseudo-code is presented at the beginning of the supplementary document (see Algorithm 1).

The first part of the algorithm can be interpreted as an initialization for its second part, and consists in applying a spectral decomposition of a $n \times n$ random matrix $A$ constructed from the observed labels. More precisely, $A = \sum_{\ell=1}^L w_\ell A^\ell$, where $A^\ell$ is the binary matrix identifying the item pairs with observed label $\ell$, i.e., for all $v, w \in \mathcal{V}$, $A_{vw}^\ell = 1$ if and only if $(v, w)$ has label $\ell$. The weight $w_\ell$ for label $\ell \in \{1, \dots, L\}$ is generated uniformly at random in $[0, 1]$, independently of other weights. From the spectral decomposition of $A$, we estimate the number of communities and provide asymptotically accurate estimates $S_1, \dots, S_K$ of the hidden clusters asymptotically accurately, i.e., we show that when $n\bar{p} = \omega(1)$, with high probability, $\hat{K} = K$ and there exists a permutation $\gamma$ of $\{1, \dots, K\}$ such that $\frac{1}{n} \left| \cup_{k=1}^K \mathcal{V}_k \setminus S_{\gamma(k)} \right| = O\left( \frac{\log(n\bar{p})^2}{n\bar{p}} \right)$. This first part of the SP algorithm is adapted from algorithms proposed for the standard SBM in [4, 22] to handle the additional labels in the model without the knowledge of the number $K$ of clusters.

The second part is novel, and is critical to ensure the optimality of the SP algorithm. It consists in first constructing an estimate $\hat{p}$ of the true parameters $p$ of the model from the matrices $(A^\ell)_{1 \leq \ell \leq L}$ and the estimated clusters $S_1, \dots, S_K$ provided in the first part of SP. We expect $p$ to be well estimated since $S_1, \dots, S_K$ are asymptotically accurate. Then our cluster estimates are iteratively improved. We run $\lfloor \log(n) \rfloor$ iterations. Let $S_1^{(t)}, \dots, S_K^{(t)}$ denote the clusters estimated after the $t$-th iteration, initialized with $(S_1^{(0)}, \dots, S_K^{(0)}) = (S_1, \dots, S_K)$. The improved clusters $S_1^{(t+1)}, \dots, S_K^{(t+1)}$ are obtained by assigning each item $v \in \mathcal{V}$ to the cluster maximizing a log-likelihood formed from $\hat{p}, S_1^{(t)}, \dots, S_K^{(t)}$, and the observations $(A^\ell)_{1 \leq \ell \leq L}$: $v$ is assigned to $S_{k^\star}^{(t+1)}$ where $k^\star = \arg\max_k \{ \sum_{i=1}^K \sum_{w \in S_i^{(t-1)}} \sum_{\ell=0}^L A_{vw}^\ell \log \hat{p}(k, i, \ell) \}$.

**Part 1: Spectral Decomposition.** The spectral decomposition is described in Lines 1 to 4 in Algorithm 1. As usual in spectral methods, the matrix $A$ is first trimmed (to remove lines and columns corresponding to items with too many observed labels – as they would perturb the spectral analysis). To this aim, we estimate the average number of labels per item, and use this estimate, denoted by $\tilde{p}$ in Algorithm 1, as a reference for the trimming process. $\Gamma$ and $A_\Gamma$ denote the set of remaining items after trimming, and the corresponding trimmed matrix, respectively.

If the number of clusters $K$ is known and if we do not account for time complexity, the two step algorithm in [4] can extract the clusters from $A_\Gamma$: first the optimal rank-$K$ approximation $A^{(K)}$ of $A_\Gamma$ is derived using the SVD; then, one applies the $k$-mean algorithm to the columns of $A^{(K)}$ to reconstruct the clusters. The number of misclassified items after this two step algorithm is obtained as follows. Let $M^\ell = \mathbb{E}[A_\Gamma^\ell]$, and $M = \sum_{\ell=1}^L w_\ell M^\ell$ (using the same weights as those defining $A$). Then, $M$ is of rank $K$. If $v$ and $w$ are in the same cluster, $M_v = M_w$ and if $v$ and $w$ do not belong to the same cluster, from (A2), we must have with high probability: $\|M_v - M_w\|_2 = \Omega(\bar{p}\sqrt{n})$. Thus, the $k$-mean algorithm misclassifies $v$ only if $\|A_v^{(K)} - M_v\|_2 = \Omega(\bar{p}\sqrt{n})$. By leveraging elements of random graph and random matrix theories, we can establish that $\sum_v \|A_v^{(k)} - M_v\|_2^2 = \|A^{(k)} - M\|_F^2 = O(n\bar{p})$ with high probability. Hence the algorithm misclassifies $O(1/\bar{p})$ items with high probability.

Here the number of clusters $K$ is not given a-priori. In this scenario, Algorithm 2 estimates the rank of $M$ using a singular value thresholding procedure. To reduce the complexity of the algorithm, the singular values and singular vectors are obtained using the iterative power method instead of a direct SVD. It is known from [10] that with $\Theta(\log(n))$ iterations, the iterative power method find singular values and the rank-$K$ approximation very accurately. Hence, when $n\bar{p} = \omega(1)$, we can easily

estimate the rank of $M$ by looking at the number of singular values above the threshold $\sqrt{n\tilde{p}}\log(n\tilde{p})$, since we know from random matrix theory that the $(K+1)$-th singular value of $A_\Gamma$ is much less than $\sqrt{n\tilde{p}}\log(n\tilde{p})$ with high probability. In the pseudo-code of Algorithm 2, the estimated rank of $M$ is denoted by $\tilde{K}$.

The rank-$\tilde{K}$ approximation of $A_\Gamma$ obtained by the iterative power method is $\hat{A} = \hat{U}\hat{V} = \hat{U}\hat{U}^\top A_\Gamma$. From the columns of $\hat{A}$, we can estimate the number of clusters and classify items. Almost every column of $\hat{A}$ is located around the corresponding column of $M$ within a distance $\frac{1}{2}\sqrt{\frac{n\tilde{p}^2}{\log(n\tilde{p})}}$, since $\sum_v \|\hat{A}_v - M_v\|_2^2 = \|\hat{A} - M\|_F^2 = O(n\bar{p}\log(n\bar{p})^2)$ with high probability (we rigorously analyze this distance in the supplementary material Section D.2). From this observation, the columns can be categorised into $K$ groups. To find these groups, we randomly pick $\log(n)$ reference columns and for each reference column, search all columns within distance $\sqrt{\frac{n\tilde{p}^2}{\log(n\tilde{p})}}$. Then, with high probability, each cluster has at least one reference column and each reference column can find most of its cluster members. Finally, the $K$ groups are identified using the reference columns. To this aim, we compute the distance of $n\log(n)$ column pairs $\hat{A}_v$, $\hat{A}_w$. Observe that $\|\hat{A}_v - \hat{A}_w\|_2 = \|\hat{V}_v - \hat{V}_w\|_2$ for any $u, v \in \Gamma$, since the columns of $\hat{U}$ are orthonormal. Now $\hat{V}_v$ is of dimension $\tilde{K}$, and hence we can identify the groups using $O(n\tilde{K}\log(n))$ operations.

**Theorem 6** *Assume that (A1) and (A2) hold, and that $n\bar{p} = \omega(1)$. After Step 4 (spectral decomposition) in the SP algorithm, with high probability, $\hat{K} = K$ and there exists a permutation $\gamma$ of $\{1, \ldots, K\}$ such that:* $\left| \cup_{k=1}^K \mathcal{V}_k \setminus S_{\gamma(k)} \right| = O\left( \frac{\log(n\bar{p})^2}{\bar{p}} \right).$

**Part 2: Successive clusters improvements.** Part 2 of the SP algorithm is described in Lines 5 and 6 in Algorithm 1. To analyze the performance of each improvement iteration, we introduce the set of items $H$ as the largest subset of $\mathcal{V}$ such that for all $v \in H$: (H1) $e(v, \mathcal{V}) \leq 10\eta n\bar{p}L$; (H2) when $v \in \mathcal{V}_k$, $\sum_{i=1}^K \sum_{\ell=0}^L e(v, \mathcal{V}_i, \ell) \log \frac{p(k,i,\ell)}{p(j,i,\ell)} \geq \frac{n\bar{p}}{\log(n\bar{p})^4}$ for all $j \neq k$; (H3) $e(v, \mathcal{V} \setminus H) \leq 2\log(n\bar{p})^2$, where for any $S \subset \mathcal{V}$ and $\ell$, $e(v, S, \ell) = \sum_{w \in S} A_{vw}^\ell$, and $e(v, S) = \sum_{\ell=1}^L e(v, S, \ell)$. Condition (H1) means that there are not too many observed labels $\ell \geq 1$ on pairs including $v$, (H2) means that an item $v \in \mathcal{V}_k$ must be classified to $\mathcal{V}_k$ when considering the log-likelihood, and (H3) states that $v$ does not share too many labels with items outside $H$.

We then prove that $|\mathcal{V} \setminus H| \leq s$ with high probability when $nD(\alpha, p) - \frac{n\bar{p}}{\log(n\bar{p})^3} \geq \log(n/s) + \sqrt{\log(n/s)}$. This is mainly done using concentration arguments to relate the quantity $\sum_{i=1}^K \sum_{\ell=0}^L e(v, \mathcal{V}_i, \ell) \log \frac{p(k,i,\ell)}{p(j,i,\ell)}$ involved in (H2) to $nD(\alpha, p)$.

Finally, we establish that if the clusters provided after the first part of the SP algorithm are asymptotically accurate, then after $\log(n)$ improvement iterations, there is no misclassified items in $H$. To that aim, we denote by $\mathcal{E}^{(t)}$ the set of misclassified items after the $t$-th iteration, and show that with high probability, for all $t$, $\frac{|\mathcal{E}^{(t+1)} \cap H|}{|\mathcal{E}^{(t)} \cap H|} \leq \frac{1}{\sqrt{n\bar{p}}}$. This completes the proof of Theorem 2, since after $\log(n)$ iterations, the only misclassified items are those in $\mathcal{V} \setminus H$.

### Acknowledgments

We gratefully acknowledge the support of the U.S. Department of Energy through the LANL/LDRD Program for this work.

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
