[Supplementary Material · labeled-sbm.pdf]

# Supplementary Material:

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

First, we guess the scaling of the lower bound of expected number of misclassified items under any algorithm, i.e., we identify its connection to the divergence $D(\alpha, p)$.

Second, we find a model $\Psi$ such that $\mathbb{E}_\Psi[\mathcal{Q}]$ approximates $D(\alpha, p)$ as $n$ grows large. To that aim, we build $\Psi$ from $\Phi$ using coupling techniques [17]. This (miraculous) coupling is inspired by the reasoning used in the first step to identify the optimal recovery rate, and is the corner-stone of the proof.

Putting the arguments together, we show that for any clustering algorithm $\pi$, $\log(n/\mathbb{E}_\Phi[\varepsilon^\pi(n)])$ must be smaller than $D(\alpha, p)$ as $n$ grows large, which gives Theorem 1. Next, we describe the two steps of the proof as well as the analysis of $\mathcal{Q}$ in more details.

## 3.1 Guessing the optimal recovery rate

Consider a LSBM with parameters $(\alpha, p)$. The optimal recovery rate is obtained using the following heuristic argument. Assume that $(\alpha, p)$ are known and that all items have been already correctly classified except for $v \in \mathcal{V}_i$. Then, applying the maximum a posteriori probability (MAP) estimator constitutes the best way of classifying $v$. Let $e(v, \mathcal{V}_k, \ell)$ denote the number of item pairs $(v, w)$ such that $w \in \mathcal{V}_k$ and with observed label $\ell$. Further introduce $\mu(v, \mathcal{V}_k) = [e(v, \mathcal{V}_k, \ell)/|\mathcal{V}_k|]_{0 \le \ell \le L}$ as the empirical probability vector defined by the label densities between $v$ and $\mathcal{V}_k$. Under the MAP estimator, $v$ is misclassified and assigned to $\mathcal{V}_j$ when

$$
\begin{aligned}
0 \quad > \quad & \log\left(\frac{\alpha_i}{\alpha_j}\right) + \sum_{k=1}^{K} \sum_{\ell=0}^{L} e(v, \mathcal{V}_k, \ell) \log\left(\frac{p(i, k, \ell)}{p(j, k, \ell)}\right) \\
= \quad & \log\left(\frac{\alpha_i}{\alpha_j}\right) + \sum_{k=1}^{K} \sum_{\ell=0}^{L} |\mathcal{V}_k| \frac{e(v, \mathcal{V}_k, \ell)}{|\mathcal{V}_k|} \left(\log\left(\frac{e(v, \mathcal{V}_k, \ell)/|\mathcal{V}_k|}{p(j, k, \ell)}\right) - \log\left(\frac{e(v, \mathcal{V}_k, \ell)/|\mathcal{V}_k|}{p(i, k, \ell)}\right)\right) \\
= \quad & \log\left(\frac{\alpha_i}{\alpha_j}\right) + \sum_{k=1}^{K} |\mathcal{V}_k| \left(KL\left(\mu(v, \mathcal{V}_k), p(j, k)\right) - KL\left(\mu(v, \mathcal{V}_k), p(i, k)\right)\right).
\end{aligned} \tag{2}
$$

Observe that the term $\log(\alpha_i/\alpha_j)$ can be neglected in the r.h.s. of (2) as $n$ grows large. Hence, from the definition of $D_{L+}(\alpha, p(i), p(j))$, $v$ is misclassified to $\mathcal{V}_j$ (i.e., (2) holds) when the following event occurs:

$$
\sum_{k=1}^{K} |\mathcal{V}_k| KL\left(\mu(v, \mathcal{V}_k), p(i, k)\right) \ge n D_{L+}(\alpha, p(i), p(j)).
$$

We can evaluate the probability of this event in the particular case when $\bar{p} = o(\frac{1}{\sqrt{n}})$ using the following claim.

**Claim 6** *When $|\mathcal{V}_k| = \Omega(n)$, $\bar{p} = o(\frac{1}{\sqrt{n}})$, $n\bar{p} = \omega(1)$, and $D = \Omega(\bar{p})$,*

$\log\left(\mathbb{P}\left\{\sum_{k=1}^{K} |\mathcal{V}_k| KL\left(\mu(v, \mathcal{V}_k), p(i, k)\right) \ge n D\right\}\right) \overset{n \to \infty}{\sim} -n D.$

Now by choosing $(i, j) = (i^\star, j^\star) = \arg\min_{i,j:i<j} D_{L+}(p(i), p(j))$, we can expect, from the previous claim applied to $D = D(\alpha, n)$, that the probability of misclassifying $v \in \mathcal{V}_{i^\star}$ scales as $\exp(-D(\alpha, n))$. Since $|\mathcal{V}_{i^\star}|$ grows linearly with $n$, this implies that the expected

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

SP algorithm consists in two parts. In the first part, corresponding to Lines 1-4 in the pseudo-code, we apply a spectral decomposition of the matrix $A = \sum_{\ell=1}^{L} w_\ell A^\ell$ constructed from the observed labels. This matrix is first trimmed, and then treated by applying the spectral decomposition algorithm, whose pseudo-code is presented in Algorithm 2. The second part of the SP algorithm, corresponding to Lines 5 and 6 in Algorithm 1, consists in improving the clusters initially identified in the first step.

---

**Algorithm 1** Spectral Partition

---

**Input:** Observation matrices $A^\ell$ for every label $\ell$ ($A_{uv}^\ell = 1$ if $\ell$ is observed between $u$ and $v$).

**1. Estimated average degree.** $\tilde{p} \leftarrow \frac{\sum_{\ell=1}^{L} \sum_{u,v} A_{uv}^\ell}{n(n-1)}$

**2. Random Weights.** $A \leftarrow \sum_{\ell=1}^{L} w_\ell A^\ell$ where the weights $w_\ell$'s are i.i.d and uniformly distributed on $[0,1]$.

**3. Trimming.** Construct $A_\Gamma = (A_{vw})_{v,w \in \Gamma}$, where $\Gamma$ is the set of nodes obtained after removing $\lfloor n \exp(-n\tilde{p}) \rfloor$ nodes having the largest $\sum_\ell \sum_{w \in V} A_{vw}^\ell$.

**4. Spectral Decomposition.** Run Algorithm 2 with input $A_\Gamma, \tilde{p}$, and output $(S_k)_{k=1,\dots,\hat{K}}$.

**5. Estimated parameters.** $\hat{p}(i,j,\ell) \leftarrow \frac{\sum_{u \in S_i} \sum_{v \in S_j} A_{uv}^\ell}{|S_i||S_j|}$ for all $1 \leq i, j \leq \hat{K}$ and $0 \leq \ell \leq L$.

**6. Improvement.**
$S_k^{(0)} \leftarrow S_k$, for all $k$
**for** $t = 1$ **to** $\log n$ **do**
    $S_k^{(t)} \leftarrow \emptyset$, for all $k$
    **for** $v \in \mathcal{V}$ **do**
        Find $k^\star = \arg\max_{1 \leq k \leq \hat{K}} \{\sum_{i=1}^{\hat{K}} \sum_{w \in S_i^{(t-1)}} \sum_{\ell=0}^{L} A_{vw}^\ell \log \hat{p}(k,i,\ell)\}$ (tie broken uniformly at random)
        $S_{k^\star}^{(t)} \leftarrow S_{k^\star}^{(t)} \cup \{v\}$
    **end for**
**end for**
$\hat{\mathcal{V}}_k \leftarrow S_k^{(\log n)}$, for all $k$
**Output:** $(\hat{\mathcal{V}}_k)_{k=1,\dots,\hat{K}}$.

---

The first part of the algorithm can be interpreted as an initialization for its second part, and consists in applying a spectral decomposition of a $n \times n$ random matrix $A$ constructed from the observed labels. More precisely, $A = \sum_{\ell=1}^{L} w_\ell A^\ell$, where $A^\ell$ is the binary matrix identifying the item pairs with observed label $\ell$, i.e., for all $v, w \in \mathcal{V}$, $A_{vw}^\ell = 1$ if and only if $(v,w)$ has label $\ell$. The weight $w_\ell$ for label $\ell \in \{1, \dots, L\}$ is generated uniformly at random in $[0,1]$, independently of other weights. From the spectral decomposition of $A$, we estimate the number of communities and provide asymptotically accurate estimates $S_1, \dots, S_K$ of the hidden clusters asymptotically accurately, i.e., we show that when $n\bar{p} = \omega(1)$, with high probability, $\hat{K} = K$ and there exists a permutation $\gamma$ of $\{1, \dots, K\}$ such that $\frac{1}{n}\left|\cup_{k=1}^{K} \mathcal{V}_k \setminus S_{\gamma(k)}\right| = O\left(\frac{\log(n\bar{p})^2}{n\bar{p}}\right)$. This first part of the SP algorithm is adapted from algorithms proposed for the standard SBM in [4, 25] to handle the additional labels in the model without the knowledge of the number $K$ of clusters.

The second part is novel, and is critical to ensure the optimality of the SP algorithm. It consists in first constructing an estimate $\hat{p}$ of the true parameters $p$ of the model from the matrices $(A^\ell)_{1 \leq \ell \leq L}$ and the estimated clusters $S_1, \dots, S_K$ provided in the first part of SP. We expect $p$ to be well estimated since $S_1, \dots, S_K$ are asymptotically accurate. Then our cluster estimates are iteratively improved. We run $\lfloor \log(n) \rfloor$ iterations. Let $S_1^{(t)}, \dots, S_K^{(t)}$ denote the clusters estimated after the $t$-th iteration, initialized with $(S_1^{(0)}, \dots, S_K^{(0)}) = (S_1, \dots, S_K)$. The improved clusters $S_1^{(t+1)}, \dots, S_K^{(t+1)}$ are obtained by assigning each item $v \in \mathcal{V}$ to the cluster maximizing a log-likelihood formed from $\hat{p}, S_1^{(t)}, \dots, S_K^{(t)}$, and the observations $(A^\ell)_{1 \leq \ell \leq L}$: $v$ is assigned to $S_{k^\star}^{(t+1)}$ where $k^\star = \arg\max_k \{\sum_{i=1}^{K} \sum_{w \in S_i^{(t-1)}} \sum_{\ell=0}^{L} A_{vw}^\ell \log \hat{p}(k,i,\ell)\}$.

**Part 1: Spectral Decomposition.** The spectral decomposition is described in Lines 1 to 4 in Algorithm 1. As usual in spectral methods, the matrix $A$ is first trimmed (to remove lines and columns

---

**Algorithm 2** Spectral decomposition
---

**Input:** $A_\Gamma, \tilde{p}$

**1. Iterative Power Method with singular value thresholding**

(Initialization) $\chi \leftarrow n$, $k \leftarrow 0$, and $\hat{U} \leftarrow 0^{n \times 1}$

**while** $\chi \geq \sqrt{n\tilde{p}} \log(n\tilde{p})$ **do**

    $k \leftarrow k + 1$,      $U_0 \leftarrow n \times 1$ Gaussian random vector

    (Iterative power method) $U_t \leftarrow (A_\Gamma)^{\lceil 2 \log(n) \rceil} U_0$

    (Orthonormalizing $U_t$) $\hat{U}_k \leftarrow \dfrac{U_t - \hat{U}_{1:k-1}(\hat{U}_{1:k-1}^\top U_t)}{\|U_t - \hat{U}_{1:k-1}(\hat{U}_{1:k-1}^\top U_t)\|_2}$

    (Estimating the $k$-th singular value) $\chi \leftarrow \|A_\Gamma \hat{U}_k\|_2$

**end while**

$\tilde{K} \leftarrow k - 1$,      $\hat{V} \leftarrow \hat{U}_{1:\tilde{K}}^\top A_\Gamma$

**2. Clustering**

$\mathcal{V}_R \leftarrow$ a subset of $\Gamma$ obtained by randomly selecting $\lceil \log(n) \rceil$ items of $\Gamma$

$Q_v \leftarrow \{w \in \Gamma : \|\hat{V}_w - \hat{V}_v\|_2^2 \leq \frac{n\tilde{p}^2}{\log(n\tilde{p})}\}$ for all $v \in \mathcal{V}_R$

(Initialization) $S_0 \leftarrow \emptyset$,      $k \leftarrow 0$, and      $\rho \leftarrow |\Gamma|$

**while** $\rho \geq \frac{\log(n\tilde{p})^4}{\tilde{p}}$ **do**

    $k \leftarrow k + 1$,      $v_k^\star \leftarrow \arg\max_{v \in \mathcal{V}_R} |Q_v \setminus \bigcup_{l=0}^{k-1} S_l|$,      $S_k \leftarrow Q_{v_k^\star} \setminus \bigcup_{l=0}^{k-1} S_l$ and $\rho \leftarrow |S_k|$.

**end while**

$\hat{K} \leftarrow k - 1$

**for** $v \in \Gamma \setminus \bigcup_{k=1}^{\hat{K}} S_k$ **do**

    $k_\star \leftarrow \arg\min_k \|\hat{V}_{v_k^\star} - \hat{V}_v\|_2$,      $S_{k_\star} \leftarrow S_{k_\star} \cup \{v\}$

**end for**

**Output:** $(S_k)_{k=1,\ldots,\hat{K}}$.

---

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

# A   Properties of the divergence $D(\alpha, p)$ and related quantities

In this section, we prove the two claims of Section 2, as well as other results on the divergence $D(\alpha, p)$ that will be instrumental in the proofs of Theorems.

## A.1   Proof of Claim 4

$D_{L+}(p(i), p(j))$ is the minimum of the objective function of the following convex optimization problem:

$$
\min_{y \in \mathcal{P}^{K \times (L+1)}} \sum_{k=1}^{K} \alpha_k \left( \sum_{\ell=1}^{L} y(k, \ell) \log \left( \frac{y(k, \ell)}{p(i, k, \ell)} \right) + (1 - \sum_{\ell=1}^{L} y(k, \ell)) \log \left( \frac{1 - \sum_{\ell=1}^{L} y(k, \ell)}{1 - \sum_{\ell=1}^{L} p(i, k, \ell)} \right) \right)
$$

$$
\text{s.t.} \quad \sum_{k=1}^{K} \alpha_k KL(y(k), p(i, k)) \geq \sum_{k=1}^{K} \alpha_k KL(y(k), p(j, k)).
$$

$$(6)$$

Note that we define $y(k, 0) = 1 - \sum_{\ell=1}^{L} y(k, \ell)$ for all $k$. Since $\bar{p} = o(1)$, one can easily check that the solution of (6) has to be $\sum_{\ell=1}^{L} y(k, \ell) = o(1)$ for all $k$. The objective function converges to infinity when $\sum_{\ell=1}^{L} y(k, \ell) = \Omega(1)$, while it has $o(\bar{p})$ when $y(k, \ell) = p(j, k, \ell)$ for all $k$ and $\ell$. Thus, we consider $\sum_{\ell=1}^{L} y(k, \ell) = o(1)$. The associated Lagrangian is:

$$
g(y, \lambda) = \sum_{k=1}^{K} \alpha_k \left( \sum_{\ell=1}^{L} y(k, \ell) \log \left( \frac{y(k, \ell)}{p(i, k, \ell)} \right) + (1 - \sum_{\ell=1}^{L} y(k, \ell)) \log \left( \frac{1 - \sum_{\ell=1}^{L} y(k, \ell)}{1 - \sum_{\ell=1}^{L} p(i, k, \ell)} \right) \right) +
$$

$$
\sum_{k=1}^{K} \alpha_k \lambda \left( \sum_{\ell=1}^{L} y(k, \ell) \log \left( \frac{p(i, k, \ell)}{p(j, k, \ell)} \right) + (1 - \sum_{\ell=1}^{L} y(k, \ell)) \log \left( \frac{1 - \sum_{\ell=1}^{L} p(i, k, \ell)}{1 - \sum_{\ell=1}^{L} p(j, k, \ell)} \right) \right).
$$

$$(7)$$

The derivative of $g(y, \lambda)$ w.r.t. $y(k, \ell)$ is computed as follows:

$$
\frac{\partial g(y, \lambda)}{\partial y(k, \ell)} = \alpha_k \left( \log \left( \frac{y(k, \ell)}{p(i, k, \ell)} \right) - \log \left( \frac{1 - \sum_{m=1}^{L} y(k, m)}{1 - \sum_{m=1}^{L} p(i, k, m)} \right) \right) +
$$

$$
\alpha_k \lambda \left( \log \left( \frac{p(i, k, \ell)}{p(j, k, \ell)} \right) - \log \left( \frac{1 - \sum_{m=1}^{L} p(i, k, m)}{1 - \sum_{m=1}^{L} p(j, k, m)} \right) \right).
$$

Observe that, since (A1) holds, $\bar{p} = o(1)$ and $\sum_{\ell=1}^{L} y(k, \ell) = o(1)$, as $n$ grows large, $\log \left( \frac{1 - \sum_{m=1}^{L} y(k, m)}{1 - \sum_{m=1}^{L} p(i, k, m)} \right)$ and $\log \left( \frac{1 - \sum_{m=1}^{L} p(i, k, m)}{1 - \sum_{m=1}^{L} p(j, k, m)} \right)$ converges to 0. Thus, (7) is minimized at

$$
y(k, \ell) = p(i, k, \ell) \left( \frac{p(j, k, \ell)}{p(i, k, \ell)} \right)^{\lambda} (1 + o(1)).
$$

$$(8)$$

When we put (8) onto (7) and use the approximation $\lim_{x \to 0} \log(1 + x) = x$ (again using $\bar{p} = o(1)$),

$$
\min_{y \in \mathcal{P}^{K \times \{0,1\}}} g(y, \lambda)
$$

$$
= \min_{y \in \mathcal{P}^{K \times \{0,1\}}} \sum_{k=1}^{K} \sum_{\ell=1}^{L} \alpha_k \left( o(\bar{p}) + \right.
$$

$$
(1 - \sum_{\ell=1}^{L} y(k, \ell)) \log \left( \frac{1 - \sum_{\ell=1}^{L} y(k, \ell)}{1 - \sum_{\ell=1}^{L} p(i, k, \ell)} \right) \left( \frac{1 - \sum_{\ell=1}^{L} p(i, k, \ell)}{1 - \sum_{\ell=1}^{L} p(j, k, \ell)} \right)^{\lambda} \right)
$$

$$
= \min_{y \in \mathcal{P}^{K \times \{0,1\}}} \sum_{k=1}^{K} \sum_{\ell=1}^{L} \alpha_k \left( o(\bar{p}) - \right.
$$

$$\sum_{\ell=1}^{L} y(k,\ell)(1+o(1)) + (1-\lambda)\sum_{\ell=1}^{L} p(i,k,\ell)(1+o(1)) + \lambda \sum_{\ell=1}^{L} p(j,k,\ell)(1+o(1)) \Bigg).$$

Therefore, the minimum value of (6) is equivalent to

$$\max_{\lambda \in [0,1]} \sum_{k=1}^{K}\sum_{\ell=1}^{L} \alpha_k \left( (1-\lambda)p(i,k,\ell) + \lambda p(j,k,\ell) - p(i,k,1)^{1-\lambda} p(j,k,\ell)^{\lambda} \right) + o(\bar{p}).$$

## A.2 Proof of Claim 5

When $\bar{p} = o(1)$, for all $i \neq j$, $\alpha_i = \frac{1}{K}$, $p(i,i,\ell) = p(\ell)$, and $p(i,j,\ell) = q(\ell)$, from Claim 4,

$$
\begin{aligned}
D_{L+}(\alpha, p(i), p(j)) &= \max_{\lambda \in [0,1]} \sum_{k=1}^{K}\sum_{\ell=1}^{L} \alpha_k \left( (1-\lambda)p(i,k,\ell) + \lambda p(j,k,\ell) - p(i,k,\ell)^{1-\lambda} p(j,k,\ell)^{\lambda} \right) \\
&= \frac{1}{K} \max_{\lambda \in [0,1]} \sum_{\ell=1}^{L} \left( p(\ell) + q(\ell) - p(\ell)^{1-\lambda} q(\ell)^{\lambda} - p(\ell)^{\lambda} q(\ell)^{1-\lambda} \right) \\
&= \frac{1}{K} \sum_{\ell=1}^{L} \left( p(\ell) + q(\ell) - 2\sqrt{p(\ell)q(\ell)} \right). \qquad (9)
\end{aligned}
$$

Now, since $\sqrt{1+x} = 1 + \frac{x}{2}(1+o(1))$ and $\log(1+x) = x(1+o(1))$ when $x = o(1)$,

$$
\begin{aligned}
-\frac{2}{K} \log \left( \sum_{\ell=0}^{L} \sqrt{p(\ell)q(\ell)} \right) &= -\frac{2}{K} \log \left( \sqrt{p(0)q(0)} + \sum_{\ell=1}^{L} \sqrt{p(\ell)q(\ell)} \right) \\
&= -\frac{2}{K} \log \left( 1 - \frac{\sum_{\ell=1}^{L} p(\ell)+q(\ell)}{2}(1+o(1)) + \sum_{\ell=1}^{L} \sqrt{p(\ell)q(\ell)} \right) \\
&= \frac{2}{K} \left( \frac{\sum_{\ell=1}^{L} p(\ell)+q(\ell)}{2} - \sum_{\ell=1}^{L} \sqrt{p(\ell)q(\ell)} \right)(1+o(1)). \qquad (10)
\end{aligned}
$$

The claim follows from (9) and (10).

## A.3 Other properties

**Lemma 8** *Let* $(i^\star, j^\star) = \arg\min_{i,j} D_{L+}(p(i), p(j))$ *and* $i^\star < j^\star$. *Then, there exists* $q \in \mathcal{P}^{K \times (L+1)}$ *such that*

$$D(\alpha, p) = \sum_{k=1}^{K} \alpha_k KL(q(k), p(i^\star, k)) = \sum_{k=1}^{K} \alpha_k KL(q(k), p(j^\star, k)).$$

*Proof.* We check by contradiction that such a $q$ exists. Indeed, assume that

$$D(\alpha, p) = \sum_{k=1}^{K} \alpha_k KL(q(k), p(i^\star, k)) > \sum_{k=1}^{K} \alpha_k KL(q(k), p(j^\star, k)).$$

Then there exists $k_0$ such that $KL(q(k_0), p(i^\star, k_0)) > KL(q(k_0), p(j^\star, k_0))$. Observe that by positivity of the $KL$ divergence, $q(k_0) \neq p(i^\star, k_0)$. Hence by continuity of the $KL$ divergence, we can construct $q'$ such that $q(k) = q'(k)$ for all $k \neq k_0$, and such that: $KL(q(k_0), p(i^\star, k_0)) - \epsilon < KL(q'(k_0), p(i^\star, k_0)) < KL(q(k_0), p(i^\star, k_0))$ and $KL(q'(k_0), p(j^\star, k_0)) < KL(q(k_0), p(j^\star, k_0)) + \epsilon$ for some $0 < \epsilon < (KL(q(k_0), p(i^\star, k_0)) - KL(q(k_0), p(j^\star, k_0)))/2$. With this choice of $q'$, we get:

$$D(\alpha, p) > \sum_{k=1}^{K} \alpha_k KL(q'(k), p(i^\star, k)) > \sum_{k=1}^{K} \alpha_k KL(q'(k), p(j^\star, k)),$$

which contradicts the definition of $D(\alpha, p)$. ∎

**Lemma 9** *When $\bar{p} = o(1)$,*

$$\lim_{n \to \infty} \frac{D(\alpha, p)}{\sum_{k=1}^{K} \frac{\alpha_k}{2} \left( \sum_{\ell=1}^{L} (\sqrt{p(i^\star, k, \ell)} - \sqrt{p(j^\star, k, \ell)})^2 \right)} \geq 1.$$

*Proof.* Let $(i^\star, j^\star) = \arg\min_{i,j} D_{L+}(\alpha, p(i), p(j))$ and $i^\star < j^\star$. From Lemma 8, there exists $q$ satisfying that

$$D(\alpha, p) = \sum_{k=1}^{K} \alpha_k KL(q(k), p(i^\star, k)) = \sum_{k=1}^{K} \alpha_k KL(q(k), p(j^\star, k)).$$

Then,

$$
\begin{aligned}
nD(\alpha, p) &= n \frac{\sum_{k=1}^{K} \left( \alpha_k KL(q(k), p(i^\star, k)) + \alpha_k KL(q(k), p(j^\star, k)) \right)}{2} \\
&= -n \sum_{k=1}^{K} \alpha_k \sum_{\ell=0}^{L} q(k, \ell) \log \left( \frac{\sqrt{p(i^\star, k, \ell) p(j^\star, k, \ell)}}{q(k, \ell)} \right) \\
&\geq n \sum_{k=1}^{K} \alpha_k \sum_{\ell=0}^{L} \left( q(k, \ell) - \sqrt{p(i^\star, k, \ell) p(j^\star, k, \ell)} \right) \\
&= n \sum_{k=1}^{K} \alpha_k \left( \frac{\sum_{\ell=1}^{L} (p(i^\star, k, \ell) + p(j^\star, k, \ell))}{2} - \sum_{\ell=1}^{L} \sqrt{p(i^\star, k, \ell) p(j^\star, k, \ell)} \right) (1 - o(1)) \\
&= n \sum_{k=1}^{K} \frac{\alpha_k}{2} \left( \sum_{\ell=1}^{L} (\sqrt{p(i^\star, k, \ell)} - \sqrt{p(j^\star, k, \ell)})^2 \right) (1 - o(1)).
\end{aligned}
$$

∎

**Lemma 10** *Under condition (A1), when $\bar{p} = o(1)$, $\limsup_{n \to \infty} \frac{D(\alpha, p)}{\eta \bar{p} L} \leq 1$.*

*Proof.* From the definition of $D(\alpha, p)$, for any $i \neq j$,

$$
\begin{aligned}
D(\alpha, p) &\leq \max \left\{ \sum_{k=1}^{K} \alpha_k KL(p(i, k), p(i, k)), \sum_{k=1}^{K} \alpha_k KL(p(i, k), p(j, k)) \right\} \\
&= \sum_{k=1}^{K} \alpha_k KL(p(i, k), p(j, k)) \\
&\leq \sum_{k=1}^{K} \alpha_k \sum_{\ell=1}^{L} \frac{(p(i, k, \ell) - p(j, k, \ell))^2}{p(j, k, \ell)} (1 + o(1)) \\
&\leq \sum_{k=1}^{K} \alpha_k \sum_{\ell=1}^{L} \eta \bar{p} (1 + o(1)) \\
&= \eta \bar{p} L (1 + o(1)),
\end{aligned}
$$

where we use $\log(1 + x) = x(1 + o(1))$ when $x = o(1)$. ∎

## B  Proof of Theorem 1

The proof consists in an appropriate *change-of-measure* argument. The originality of the proof stems from the fact that the change of measures is obtained by a judicious coupling argument [17]. In the following, we refer to $\Phi$ as the true stochastic model under which all the observed random labels are generated, and denote by $\mathbb{P}_\Phi = \mathbb{P}$ (resp. $\mathbb{E}_\Phi[\cdot] = \mathbb{E}[\cdot]$) the corresponding probability measure (resp. expectation). We recall that $\Phi$ is defined by the parameters $(\alpha, p)$, and that under $\Phi$, the nodes are first attached to the various clusters according to the distribution $\alpha$, and the labels between two nodes are then generated using distributions $p$. The proof consists in constructing a perturbed stochastic model

$\Psi$ coupling the labels generated under $\Phi$ with those generated under $\Psi$. We denote by $\mathbb{P}_\Psi$ (resp. $\mathbb{E}_\Psi[\cdot] = \mathbb{E}[\cdot]$) the probability measure (resp. expectation) under the perturbed model $\Psi$. We then relate the proportion of misclassified nodes under any given clustering algorithm $\pi$ to the distribution under $\mathbb{P}_\Psi$ of a quantity $\mathcal{Q}$ that resembles the log-likelihood ratio of the observed labels under $\mathbb{P}_\Phi$ and $\mathbb{P}_\Psi$. The analysis of the likelihood ratio finally provides the desired lower bound on the expected misclassified nodes under $\pi$. Next, we detail each step of the proof.

**Coupling and the perturbed stochastic model $\Psi$.** Let $(i^\star, j^\star) = \arg\min_{i,j: i<j} D_{L+}(p(i), p(j))$, and let $v^\star$ denote the smallest node index that belongs to cluster $i^\star$ or $j^\star$. If both $\mathcal{V}_{i^\star}$ and $\mathcal{V}_{j^\star}$ are empty, we define $v^\star = n$. Let $q \in [0,1]^{K \times (L+1)}$ satisfy:

$$D(\alpha, p) = \sum_{k=1}^{K} \alpha_k KL(q(k), p(i^\star, k)) = \sum_{k=1}^{K} \alpha_k KL(q(k), p(j^\star, k)).$$

There exists such a $q$ from Lemma 8. Now to define the perturbed stochastic model $\Psi$, we couple the generation of labels under $\Phi$ and $\Psi$ as follows.

1. We first generate construct the random clusters $\mathcal{V}_1, \ldots, \mathcal{V}_K$ under $\Phi$, and extract $i^\star$, $j^\star$, and $v^\star$. The clusters generated under $\Psi$ are the same as those generated under $\Phi$. For any $v \in \mathcal{V}$, we denote by $\sigma(v)$ the cluster of node $v$.

2. For all nodes $v, w \neq v^\star$, the labels generated under $\Psi$ are the same as those generated under $\Phi$, i.e., the label $\ell$ is observed on the edge $(v, w)$ with probability $p(\sigma(v), \sigma(w), \ell)$.

3. Under $\Psi$, for any $v \neq v^\star$, the observed label on the edge $(v, v^\star)$ under $\Psi$ is $\ell$ with probability $q(\sigma(v), \ell)$.

**The log-likelihood ratio and its connection to the expected number of misclassified nodes.** Let $x_{v,w}$ denote the label observed on the edge $(v, w)$. We introduce $\mathcal{Q}$, referred to as the pseudo-log-likelihood ratio of the observed labels under $\mathbb{P}_\Phi$ and $\mathbb{P}_\Psi$) as:

$$\mathcal{Q} = \sum_{v=1}^{v^\star - 1} \log \frac{q(\sigma(v), x_{v^\star, v})}{p(\sigma(v^\star), \sigma(v), x_{v^\star, v})} + \sum_{v=v^\star+1}^{n} \log \frac{q(\sigma(v), x_{v^\star, v})}{p(\sigma(v^\star), \sigma(v), x_{v^\star, v})}. \tag{11}$$

Let $\pi$ denote a clustering algorithm with output $(\hat{\mathcal{V}}_k)_{1 \le k \le K}$, and let $\mathcal{E} = \bigcup_{1 \le k \le K} \hat{\mathcal{V}}_k \setminus \mathcal{V}_k$ be the set of misclassified nodes under $\pi$. Note that in general in our proofs, we always assume without loss of generality that $|\bigcup_{1 \le k \le K} \hat{\mathcal{V}}_k \setminus \mathcal{V}_k| \le |\bigcup_{1 \le k \le K} \hat{\mathcal{V}}_{\gamma(k)} \setminus \mathcal{V}_k|$ for any permutation $\gamma$, so that the set of misclassified nodes is really $\mathcal{E}$. We denote by $\varepsilon^\pi(n) = |\mathcal{E}|$. Since under $\Phi$, nodes are interchangeable (remember that nodes are assigned to the various clusters in an i.i.d. manner), we have:

$$n\mathbb{P}_\Phi\{v \in \mathcal{E}\} = \mathbb{E}_\Phi[\varepsilon^\pi(n)] = \mathbb{E}[\varepsilon^\pi(n)].$$

Next, we establish a relationship between $\mathbb{E}[\varepsilon^\pi(n)]$ and the distribution of $\mathcal{Q}$ under $\mathbb{P}_\Psi$. For any function $f(n)$, we have:

$$\mathbb{P}_\Psi\{\mathcal{Q} \le f(n)\} = \mathbb{P}_\Psi\{\mathcal{Q} \le f(n), v^\star \in \mathcal{E}\} + \mathbb{P}_\Psi\{\mathcal{Q} \le f(n), v^\star \notin \mathcal{E}\}. \tag{12}$$

Using $\mathcal{Q}$, we get:

$$
\begin{aligned}
\mathbb{P}_\Psi\{\mathcal{Q} \le f(n), v^\star \in \mathcal{E}\} &= \int_{\{\mathcal{Q} \le f(n), v^\star \in \mathcal{E}\}} d\mathbb{P}_\Psi \\
&= \int_{\{\mathcal{Q} \le f(n), v^\star \in \mathcal{E}\}} \exp(\mathcal{Q}) d\mathbb{P}_\Phi \\
&\le \exp(f(n)) \mathbb{P}_\Phi\{\mathcal{Q} \le f(n), v^\star \in \mathcal{E}\} \\
&\le \exp(f(n)) \mathbb{P}_\Phi\{v^\star \in \mathcal{E}\} \\
&\le \exp(f(n)) \frac{\mathbb{E}_\Phi[\varepsilon^\pi(n)]}{(\alpha_{i^\star} + \alpha_{j^\star})n},
\end{aligned} \tag{13}
$$

where the last inequality is obtained from the fact that we cannot distinguish between $v^\star$ and any other $v \in \mathcal{V}_{\sigma(v^\star)}$. Indeed,

$$\mathbb{P}_\Phi\{v^\star \in \mathcal{E}\} = \mathbb{P}_\Phi\{v \in \mathcal{E} | v \in \mathcal{V}_{i^\star} \cup \mathcal{V}_{j^\star}\}$$

$$
\begin{aligned}
&= \frac{\mathbb{P}_\Phi\{v \in \mathcal{E}, v \in \mathcal{V}_{i^\star} \cup \mathcal{V}_{j^\star}\}}{\mathbb{P}_\Phi\{v \in \mathcal{V}_{i^\star} \cup \mathcal{V}_{j^\star}\}} \\
&\leq \frac{\mathbb{P}_\Phi\{v \in \mathcal{E}\}}{\mathbb{P}_\Phi\{v \in \mathcal{V}_{i^\star} \cup \mathcal{V}_{j^\star}\}} = \frac{\mathbb{E}_\Phi[\varepsilon^\pi(n)]}{(\alpha_{i^\star} + \alpha_{j^\star})n}.
\end{aligned}
$$

Furthermore, since under the stochastic model $\Psi$, the observed labels do not depend on whether $v^\star$ belongs to cluster $i^\star$ or $j^\star$, we have:

$$
\begin{aligned}
\mathbb{P}_\Psi\{v^\star \in \hat{\mathcal{V}}_{i^\star} | v^\star \in \mathcal{V}_{i^\star}\} &= \mathbb{P}_\Psi\{v^\star \in \hat{\mathcal{V}}_{i^\star} | v^\star \in \mathcal{V}_{j^\star}\} \quad \text{and} \\
\mathbb{P}_\Psi\{v^\star \in \hat{\mathcal{V}}_{j^\star} | v^\star \in \mathcal{V}_{i^\star}\} &= \mathbb{P}_\Psi\{v^\star \in \hat{\mathcal{V}}_{j^\star} | v^\star \in \mathcal{V}_{j^\star}\}.
\end{aligned}
$$

Finally, since $\mathbb{P}_\Psi\{v^\star \in \hat{\mathcal{V}}_{i^\star} | v^\star \in \mathcal{V}_{i^\star}\} + \mathbb{P}_\Psi\{v^\star \in \hat{\mathcal{V}}_{j^\star} | v^\star \in \mathcal{V}_{i^\star}\} \leq 1$, we also have:

$$
\begin{aligned}
&\mathbb{P}_\Psi\{\mathcal{Q} \leq f(n), v^\star \notin \mathcal{E}\} \\
&\leq \mathbb{P}_\Psi\{v^\star \notin \mathcal{E}\} \\
&= \frac{\alpha_{i^\star}}{\alpha_{i^\star} + \alpha_{j^\star}} \mathbb{P}_\Psi\{v^\star \in \hat{\mathcal{V}}_{i^\star} | v^\star \in \mathcal{V}_{i^\star}\} + \frac{\alpha_{j^\star}}{\alpha_{i^\star} + \alpha_{j^\star}} \mathbb{P}_\Psi\{v^\star \in \hat{\mathcal{V}}_{j^\star} | v^\star \in \mathcal{V}_{j^\star}\} \\
&= \frac{\alpha_{i^\star}}{\alpha_{i^\star} + \alpha_{j^\star}} \mathbb{P}_\Psi\{v^\star \in \hat{\mathcal{V}}_{i^\star} | v^\star \in \mathcal{V}_{i^\star}\} + \frac{\alpha_{j^\star}}{\alpha_{i^\star} + \alpha_{j^\star}} \mathbb{P}_\Psi\{v^\star \in \hat{\mathcal{V}}_{j^\star} | v^\star \in \mathcal{V}_{i^\star}\} \\
&\leq \frac{\alpha_{j^\star}}{\alpha_{i^\star} + \alpha_{j^\star}}.
\end{aligned}
\tag{14}
$$

Combining (12), (13), and (14), we conclude that:

$$
\mathbb{P}_\Psi\{\mathcal{Q} \leq f(n)\} \leq \exp(f(n)) \frac{\mathbb{E}_\Phi[\varepsilon^\pi(n)]}{(\alpha_{i^\star} + \alpha_{j^\star})n} + \frac{\alpha_{j^\star}}{\alpha_{i^\star} + \alpha_{j^\star}}.
\tag{15}
$$

The previous equation provides the desired generic relationship between $\mathbb{E}_\Phi[\varepsilon^\pi(n)]$ and $\mathbb{P}_\Psi\{\mathcal{Q} \leq f(n)\}$ from which can deduce a necessary condition for $\mathbb{E}[\varepsilon^\pi(n)] \leq s$. Applying (15) with $f(n) = \log(n/\mathbb{E}_\Phi[\varepsilon^\pi(n)]) - \log(2/\alpha_{i^\star})$, we have:

$$
\mathbb{P}_\Psi\{\mathcal{Q} \leq \log(n/\mathbb{E}_\Phi[\varepsilon^\pi(n)]) - \log(2/\alpha_{i^\star})\} \leq 1 - \frac{\alpha_{i^\star}}{2} < 1 - \frac{\alpha_{i^\star}}{4}.
\tag{16}
$$

In addition, from Chebyshev's inequality,

$$
\mathbb{P}_\Psi\left\{\mathcal{Q} \leq \mathbb{E}_\Psi[\mathcal{Q}] + \sqrt{\frac{4}{\alpha_{i^\star}} \mathbb{E}_\Psi[(\mathcal{Q} - \mathbb{E}_\Psi[\mathcal{Q}])^2]}\right\} \geq 1 - \frac{\alpha_{i^\star}}{4}.
\tag{17}
$$

From (16) and (17), we deduce that:

$$
\log(n/\mathbb{E}_\Phi[\varepsilon^\pi(n)]) - \log(2/\alpha_{i^\star}) \leq \mathbb{E}_\Psi[\mathcal{Q}] + \sqrt{\frac{4}{\alpha_{i^\star}} \mathbb{E}_\Psi[(\mathcal{Q} - \mathbb{E}_\Psi[\mathcal{Q}])^2]},
$$

and thus, a necessary condition for $\mathbb{E}[\varepsilon^\pi(n)] \leq s$ is:

$$
\log(n/s) - \log(2/\alpha_{i^\star}) \leq \mathbb{E}_\Psi[\mathcal{Q}] + \sqrt{\frac{4}{\alpha_{i^\star}} \mathbb{E}_\Psi[(\mathcal{Q} - \mathbb{E}_\Psi[\mathcal{Q}])^2]}.
\tag{18}
$$

**Analysis of the log-likelihood ratio.** In view of (18), we can obtain a necessary condition for $\mathbb{E}[\varepsilon^\pi(n)] \leq s$ if we evaluate $\mathbb{E}_\Psi[\mathcal{Q}]$ and $\mathbb{E}_\Psi[(\mathcal{Q} - \mathbb{E}_\Psi[\mathcal{Q}])^2]$.

(i) We first compute $\mathbb{E}_\Psi[\mathcal{Q}]$. Note that in view of the definition of $v^\star$, a node whose index is smaller than $v^\star$ cannot be in $\mathcal{V}_{i^\star}$ or $\mathcal{V}_{j^\star}$, whereas a node whose index $v$ is larger than $v^\star$ can be in any cluster (and the cluster of such a $v$ is drawn according to the distribution $\alpha$ independently of other nodes). This slightly complicates the computation of the expectation of the two sums defining $\mathcal{Q}$ in (11). To circumvent this problem, we can observe that $v^\star$ is rather small, i.e., less $\log(n)^2$ with high probability, and that hence, we can approximate $\mathbb{E}_\Psi[\mathcal{Q}]$ by $\mathbb{E}_\Psi[\sum_{v=v^\star+1}^n \log \frac{q(\sigma(v), x_{v^\star, v})}{p(\sigma(v^\star), \sigma(v), x_{v^\star, v})}]$, which is itself well-approximated by $nD(\alpha, p)$. More formally, since $\mathbb{P}\{v^\star \leq m\} = 1 - (1 - \alpha_{i^\star} - \alpha_{j^\star})^m$,

$$
\mathbb{P}\{v^\star \leq \log(n)^2\} \geq 1 - \frac{1}{n^4}.
\tag{19}
$$

Hence from condition (A1), (19), and the definition of $\mathcal{Q}$,

$$
\begin{aligned}
\mathbb{E}_\Psi[\mathcal{Q}] &= \mathbb{P}\{v^\star > \log(n)^2\}\mathbb{E}_\Psi[\mathcal{Q}|v^\star > \log(n)^2] + \mathbb{P}\{v^\star \le \log(n)^2\}\mathbb{E}_\Psi[\mathcal{Q}|v^\star \le \log(n)^2] \\
&\le \frac{\log\eta}{n^3} + \mathbb{E}_\Psi[\mathcal{Q}|v^\star \le \log(n)^2] \\
&\le \frac{\log\eta}{n^3} + \mathbb{E}_\Psi\left[\sum_{v=1}^{v^\star-1}\log\frac{q(\sigma(v), x_{v^\star,v})}{p(\sigma(v^\star), \sigma(v), x_{v^\star,v})}\Big|v^\star \le \log(n)^2\right] + nD(\alpha, p) \\
&\le \frac{\log\eta}{n^3} + \mathbb{E}_\Psi\left[(v^\star-1)\sum_{k\notin\{i^\star,j^\star\}}\frac{\alpha_k KL(q(k), p(\sigma(v^\star, k)))}{1 - \alpha_{i^\star} - \alpha_{j^\star}}\Big|v^\star \le \log(n)^2\right] + nD(\alpha, p) \\
&\le \left(n + 2\log(n)^2\log\eta\right)D(\alpha, p) + \frac{\log\eta}{n^3}, \quad (20)
\end{aligned}
$$

where the last inequality stems from the fact that $2KL(q(i), p(\sigma(v^\star, i)))\log\eta \ge KL(q(j), p(\sigma(v^\star, j)))$ for all $i$ and $j$ from condition (A1).

(ii) To compute $\mathbb{E}_\Psi[(\mathcal{Q} - \mathbb{E}_\Psi[\mathcal{Q}])^2]$, we evaluate $\mathbb{E}_\Psi[(\mathcal{Q} - nD(\alpha, p))^2|\sigma(v^\star) = i^\star]$ and $\mathbb{E}_\Psi[(\mathcal{Q} - nD(\alpha, p))^2|\sigma(v^\star) = j^\star]$. From condition (A1), (19), and the definition of $\mathcal{Q}$,

$$
\begin{aligned}
&\mathbb{E}_\Psi[(\mathcal{Q} - nD(\alpha, p))^2|\sigma(v^\star) = i^\star] \\
&= \mathbb{P}\{v^\star \le \log(n)^2\}\mathbb{E}_\Psi[(\mathcal{Q} - nD(\alpha, p))^2|\sigma(v^\star) = i^\star, v^\star \le \log(n)^2] \\
&\quad + \mathbb{P}\{v^\star > \log(n)^2\}\mathbb{E}_\Psi[(\mathcal{Q} - nD(\alpha, p))^2|\sigma(v^\star) = i^\star, v^\star > \log(n)^2] \\
&\le \mathbb{E}_\Psi[(\mathcal{Q} - nD(\alpha, p))^2|\sigma(v^\star) = i^\star, v^\star \le \log(n)^2] \\
&\quad + \frac{1}{n^4}\mathbb{E}_\Psi[(\mathcal{Q} - nD(\alpha, p))^2|\sigma(v^\star) = i^\star, v^\star > \log(n)^2] \\
&= O(n\bar{p}).
\end{aligned}
$$

To derive the above inequality, we have used:

$$
\begin{aligned}
&\mathbb{E}_\Psi\left[\left(\sum_{v=v^\star+1}^n\left(\log\frac{q(\sigma(v), x_{v^\star,v})}{p(\sigma(v^\star), \sigma(v), x_{v^\star,v})} - D(\alpha, p)\right)\right)^2\Big|\sigma(v^\star) = i^\star\right] \\
&= \sum_{v=v^\star+1}^n\mathbb{E}_\Psi\left[\left(\log\frac{q(\sigma(v), x_{v^\star,v})}{p(i^\star, \sigma(v), x_{v^\star,v})} - D(\alpha, p)\right)^2\Big|\sigma(v^\star) = i^\star\right] \\
&= O(n\bar{p}) \quad \text{and} \\
&\mathbb{E}_\Psi\left[\left(\sum_{v=1}^{v^\star-1}\left(\log\frac{q(\sigma(v), x_{v^\star,v})}{p(\sigma(v^\star), \sigma(v), x_{v^\star,v})} - D(\alpha, p)\right)\right)^2\Big|\sigma(v^\star) = i^\star\right] \\
&= O(v^\star\bar{p} + (v^\star\bar{p})^2),
\end{aligned}
$$

where we use (A1) and the fact that every label is generated independently. Using the same approach, we can also conclude that $\mathbb{E}_\Psi[(\mathcal{Q} - nD(\alpha, p))^2|\sigma(v^\star) = j^\star] = O(n\bar{p})$. In summary, we have:

$$
\mathbb{E}_\Psi[(\mathcal{Q} - \mathbb{E}_\Psi[\mathcal{Q}])^2] = O(n\bar{p}). \quad (21)
$$

We are ready to complete the proof of Theorem 1. From (18), (20), (21), and Lemma 9, when the expected number of misclassified nodes is less than $s$ (i.e., $\mathbb{E}[\varepsilon^\pi(n)] \le s$), we must have:

$$
\liminf_{n\to\infty}\frac{nD(\alpha, p)}{\log(n/s)} \ge 1.
$$

∎

## C   Performance of the SP Algorithm – Proof of Theorem 2

**Notations.** We use the standard matrix norm $\|A\| = \sup\limits_{x:\|x\|_2=1}\|Ax\|_2$. We denote by $M^\ell$ the expectation of the matrix of $A^\ell$, i.e., $M^\ell_{u,v} = p(i, j, \ell)$ when $u \in \mathcal{V}_i$ and $v \in \mathcal{V}_j$. Let $M =$

$\sum_{\ell=1}^{L} w_\ell M^\ell$. We define $A_\Gamma$ to denote the adjacency matrix obtained after trimming (Step 3 in Algorithm 1). For any matrix $R \in \mathbb{R}^{n \times n}$, we define the matrix $R_\Gamma$ the square matrix formed by the lines and columns of $R$ whose indexes are in $\Gamma$. Hence, we can define $A_\Gamma^\ell$, $M_\Gamma^\ell$, and $M_\Gamma$ where $\Gamma$ is the set of items obtained after the trimming process (Line 3) in the SP algorithm (when taking the expectation to get for example $M_\Gamma$, we condition on $\Gamma$). We introduce the noise matrices $X_\Gamma^\ell = A_\Gamma^\ell - M_\Gamma^\ell$ and $X_\Gamma = \sum_{\ell=1}^{L} w_\ell X_\Gamma^\ell$. We also denote by $e(v, S, \ell) = \sum_{w \in S} A_{vw}^\ell$ the total number of item pairs with observed label $\ell$ including the item $v$ and an item from $S$ and $\mu(v, S, \ell) = \frac{e(v,S,\ell)}{|S|}$ the empirical density of label $\ell$. Let $e(v, S) = \sum_{\ell=1}^{L} e(v, S, \ell)$ and $\mu(v, S) = [\mu(v, S, \ell)]_{0 \le \ell \le L}$. In what follows, $e(v, \mathcal{V})$ is referred to as the *degree* of item $v$ (the number of observed labels different than 0 of pairs of items including $v$).

**Outline of the proof.** To analyze the performance of the SP algorithm, we first state preliminary lemmas. Lemma 11 is concerned with the concentration of the degree of the various items. Lemma 12 provides an upper bound of the matrix norm of random noise matrix $X_\Gamma^\ell$. From these two lemmas, we analyze the performance of the first part of the SP algorithm, and prove Theorem 7. To analyze the second part of the SP algorithm consisting of $\log(n)$ improvement iterations, we introduce an appropriate set of items $H$ such that that $\mathcal{V} \setminus H$ is of cardinality less than $s$ with high probability under the condition that $nD(\alpha, p) - \frac{n\bar{p}}{\log(n\bar{p})^3} \ge \log(n/s) + \sqrt{\log(n/s)}$. We further bound the rate of improvement of our cluster estimates in each iteration when restricted to the set of items $H$, and deduce that after $\log(n)$ iterations, no item in $H$ is misclassified.

## C.1 Preliminary lemmas

**Lemma 11** *For every $v \in \mathcal{V}$ and $c \ge 1$, we have*

$$\mathbb{P}\{e(v, \mathcal{V}) \ge 10cn\bar{p}L\} \le \exp(-10cn\bar{p}L).$$

*Proof.* From Markov inequality,

$$
\begin{aligned}
\mathbb{P}\{e(v, \mathcal{V}) \ge 10n\bar{p}L\} &\le \inf_{\theta > 0} \frac{\prod_{k=1}^{K} \mathbb{E}\left[\exp(\theta e(v, \mathcal{V}_k))\right]}{\exp(\theta 10cn\bar{p}L)} \\
&\le \inf_{\theta > 0} \frac{\prod_{k=1}^{K} \left(1 + \bar{p}L(\exp(\theta) - 1)\right)^{\alpha_k n}}{\exp(\theta 10cn\bar{p}L)} \\
&\le \inf_{\theta > 0} \frac{\prod_{k=1}^{K} \left(\exp(\bar{p}L(\exp(\theta) - 1))\right)^{\alpha_k n}}{\exp(\theta 10cn\bar{p}L)} \\
&\le \exp(-10cn\bar{p}L),
\end{aligned}
$$

where we derive the last inequality choosing $\theta = 2$. ∎

**Lemma 12 (Lemma 8.5 of [4])** *When $e(v, \mathcal{V}, \ell) \le \Delta$ for all $v \in \Gamma$, with high probability,*

$$\|X_\Gamma^\ell\| = O(\sqrt{n\bar{p} + \Delta}).$$

The proof of Lemma 12 relies on arguments used in the spectral analysis of random graphs, see [7] and [4].

**Lemma 13** *For all $v \in \mathcal{V}_k$ and $D \ge 0$,*

$$
\mathbb{P}\left\{ \left( \sum_{i=1}^{K} |\mathcal{V}_i| KL(\mu(v, \mathcal{V}_i), p(k, i)) \ge nD \right) \cap \left( e(v, \mathcal{V}) \le 10\eta n\bar{p}L \right) \right\}
$$

$$
\le \exp\left( -nD + KL\log(10\eta L n\bar{p}) + \frac{100\eta^2 n\bar{p}^2 L^2}{\alpha_1} \right).
$$

*Proof.* Let $\mathcal{X}$ be a set of $K \times (L+1)$ matrices such that

$$\mathcal{X} = \left\{ \boldsymbol{x} \in \mathbb{Z}^{K \times (L+1)} : \quad \sum_{i=1}^{K} \sum_{\ell=1}^{L} x_{i,\ell} \leq 10\eta n \bar{p} L, \quad \text{and} \quad \sum_{\ell=0}^{L} x_{i,\ell} = |\mathcal{V}_i| \quad \text{for all} \quad 1 \leq i \leq K \right\}.$$

For notational simplicity, we use $[\frac{x_{i,\ell}}{|\mathcal{V}_i|}]$ instead of $[\frac{x_{i,\ell}}{|\mathcal{V}_i|}]_{0 \leq \ell \leq L}$ to represent the probability mass vector on labels defined by $x_i$. With a slight abuse of notation, we denote by $e(v)$ the $K \times (L+1)$ matrix whose $(i,\ell)$ element is $e(v, \mathcal{V}_i, \ell)$. Then, for $v \in \mathcal{V}_k$,

$$\mathbb{P}\left\{ \left( \sum_{i=1}^{K} |\mathcal{V}_i| KL(\mu(v, \mathcal{V}_i), p(k, i)) \geq nD \right) \cap \left( e(v, \mathcal{V}) \leq 10 n \bar{p} L \right) \right\}$$

$$= \sum_{\boldsymbol{x} \in \mathcal{X}} \mathbb{P}\{e(v) = \boldsymbol{x}\} \mathbb{P}\left\{ \sum_{i=1}^{K} |\mathcal{V}_i| KL(\mu(v, \mathcal{V}_i), p(k, i)) \geq nD \Big| e(v) = \boldsymbol{x} \right\}$$

$$\leq \sum_{\boldsymbol{x} \in \mathcal{X}} \mathbb{P}\{e(v) = \boldsymbol{x}\} \frac{\exp\left( \sum_{i=1}^{K} |\mathcal{V}_i| KL([\frac{x_{i,\ell}}{|\mathcal{V}_i|}], p(k, i)) \right)}{\exp(nD)}$$

$$\leq \sum_{\boldsymbol{x} \in \mathcal{X}} \mathbb{P}\{e(v) = \boldsymbol{x}\} \frac{\prod_{i=1}^{K} \prod_{\ell=0}^{L} \left( \frac{x_{i,\ell}}{|\mathcal{V}_i| p(k,i,\ell)} \right)^{x_{i,\ell}}}{\exp(nD)}$$

$$\overset{(a)}{\leq} \frac{1}{\exp(nD)} \sum_{\boldsymbol{x} \in \mathcal{X}} \prod_{i=1}^{K} \left( \left( 1 - \frac{\sum_{\ell=1}^{L} x_{i,\ell}}{|\mathcal{V}_i|} \right)^{x_{i,0}} \exp(\sum_{\ell=1}^{L} x_{i,\ell}) \right)$$

$$= \frac{1}{\exp(nD)} \sum_{\boldsymbol{x} \in \mathcal{X}} \prod_{i=1}^{K} \exp\left( (|\mathcal{V}_i| - \sum_{\ell=1}^{L} x_{i,\ell}) \log\left( 1 - \frac{\sum_{\ell=1}^{L} x_{i,\ell}}{|\mathcal{V}_i|} \right) + \sum_{\ell=1}^{L} x_{i,\ell} \right)$$

$$\leq \frac{1}{\exp(nD)} \sum_{\boldsymbol{x} \in \mathcal{X}} \prod_{i=1}^{K} \exp\left( \frac{(\sum_{\ell=1}^{L} x_{k,\ell})^2}{|\mathcal{V}_i|} \right)$$

$$\leq \frac{(10\eta n \bar{p} L)^{KL} \exp(100\eta^2 n \bar{p}^2 L^2/\alpha_1)}{\exp(nD)}$$

$$= \exp\left( -nD + KL \log(10\eta Ln\bar{p}) + \frac{100\eta^2 n \bar{p}^2 L^2}{\alpha_1} \right),$$

where $(a)$ stems from the following inequality:

$$\mathbb{P}\{e(v, \mathcal{V}_i, \ell) = x_{i,\ell} \quad \text{for all} \quad i, \ell\}$$

$$\leq \prod_{i=1}^{K} \left( p(k,i,0)^{x_{i,0}} \prod_{\ell=1}^{L} \binom{|\mathcal{V}_i|}{x_{i,\ell}} p(k,i,\ell)^{x_{k,\ell}} \right)$$

$$\leq \prod_{i=1}^{K} \left( p(k,i,0)^{x_{i,0}} \prod_{\ell=1}^{L} \left( \frac{e|\mathcal{V}_i|}{x_{i,\ell}} \right)^{x_{i,\ell}} p(k,i,\ell)^{x_{i,\ell}} \right).$$

∎

## C.2 Part 1 of the SP algorithm – Proof of Theorem 7

Recall that $\hat{A} = \hat{U}\hat{V} = \hat{U}\hat{U}^\top A_\Gamma$ and $\|\hat{A}_u - \hat{A}_v\| = \|\hat{V}_u - \hat{V}_v\|$. We can bound the number of misclassified items as follows:

- with high probability, we have

$$\|\hat{A} - M_\Gamma\|_F^2 = \sum_{v \in \Gamma} \|\hat{A}_v - M_{v,\Gamma}\|_2^2 = O(n\bar{p} \log(n\bar{p})^2); \tag{22}$$

- with high probability, every item pair $u$ and $v$ satisfies that when $\sigma(v)$ represents the cluster of $v$ and $M_{v,\Gamma}$ denotes the column vector of $M_\Gamma$ on $v$,

$$\|M_{u,\Gamma} - M_{v,\Gamma}\|_2^2 = \Omega\left(n\bar{p}^2\right) \quad \text{when} \quad \sigma(u) \neq \sigma(v), \tag{23}$$

  since every $w_\ell$ is generated uniformly at random in $[0,1]$ and (A2) holds;
- (23) suggests that if $v$ is misclassified by Algorithm 2, then we should have:

$$\|\hat{A}_v - M_{v,\Gamma}\|_2^2 = \Omega\left(n\bar{p}^2\right); \tag{24}$$

- from (22) and (24), with high probability,

$$\left|\bigcup_{k=1}^{K}(V_k \setminus S_k)\right| = O\left(\frac{\log(n\bar{p})^2}{\bar{p}}\right).$$

Next, we prove (22) and (24).

*Proof of* (22). First observe that from the definition of $\Gamma$,

$$
\begin{aligned}
\mathbb{P}\left\{\max_{v\in\Gamma} e(v,\mathcal{V}) \geq 10n\bar{p}L\right\} &= \mathbb{P}\{|\{v : e(v,\mathcal{V}) \geq 10n\bar{p}L\}| > \lfloor n\exp(-n\tilde{p})\rfloor\} \\
&\leq \frac{n\exp(-10n\bar{p}L)}{\lfloor n\exp(-n\tilde{p})\rfloor + 1} \\
&\leq \exp(-5n\bar{p}L),
\end{aligned}
$$

where the first inequality stems from Lemma 11 and Markov inequality. Therefore, with high probability,

$$\max_{v\in\Gamma} e(v,\mathcal{V}) \leq 10n\bar{p}L. \tag{25}$$

When the degrees of items are bounded, the standard matrix norm of each noise matrix $X_\Gamma^\ell$ can be bounded using Lemma 12. From (25) and Lemma 12,

$$
\begin{aligned}
\|X_\Gamma\| &\leq \sum_{\ell=1}^{L} w_\ell \|X_\Gamma^\ell\| \\
&= \sum_{\ell=1}^{L} O(w_\ell\sqrt{n\bar{p} + 10n\bar{p}L}) \\
&= O(\sqrt{n\bar{p}}). \tag{26}
\end{aligned}
$$

Let $\tilde{K}$ be the number of columns of $\hat{U}$. Since $\hat{A}$ is the $\tilde{K}$-rank approximation of $A_\Gamma$ obtained by the iterative power method with $2\log(n)$ iterations, from Theorem 9.1 and Theorem 9.2 in [11], with high probability,

$$\frac{1}{2}s_k(A_\Gamma) \leq \|A_\Gamma\hat{U}_k\| \leq s_k(A_\Gamma) \quad \text{and} \quad \|A_\Gamma(I - \hat{U}_{1:k}\hat{U}_{1:k}^\top)\| \leq 2s_{k+1}(A_\Gamma). \tag{27}$$

Since $\|A_\Gamma\hat{U}_K\| \leq s_{K+1}(A_\Gamma) \leq \|X_\Gamma\| = O(\sqrt{n\bar{p}})$ from Lemma 12 and (27), $\tilde{K} \leq K$ and thus the rank of $(\hat{A} - M_\Gamma)$ is less than $2K$. Therefore,

$$
\begin{aligned}
\|\hat{A} - M_\Gamma\|_F^2 &\leq 2K\|\hat{A} - M_\Gamma\|^2 \\
&\leq 4K\left(\|\hat{A} - A_\Gamma\|^2 + \|A_\Gamma - M_\Gamma\|^2\right) \\
&\leq O(n\bar{p}\log(n\bar{p})^2), \tag{28}
\end{aligned}
$$

where the last inequality stems from the fact that $\|A_\Gamma - M_\Gamma\| = \|X_\Gamma\| = O(\sqrt{n\bar{p}})$ and $\|\hat{A} - A_\Gamma\| \leq 2s_{\tilde{K}+1}(A_\Gamma) = O(\sqrt{n\bar{p}}\log(n\bar{p}))$ from (27).

*Proof of* (24). Define the following sets:

$$
\begin{aligned}
\mathcal{I}_k &= \{v \in \mathcal{V}_k \cap \Gamma : \|\hat{A}_v - M_\Gamma^k\|^2 \leq \frac{1}{4}\frac{n\tilde{p}^2}{\log(n\tilde{p})}\} \\
\mathcal{O} &= \{v \in \Gamma : \|\hat{A}_v - M_\Gamma^k\|^2 \geq 4\frac{n\tilde{p}^2}{\log(n\tilde{p})} \quad \text{for all} \quad 1 \leq k \leq K\}.
\end{aligned}
$$

These sets are designed so that

(i) $|(\cup_{k=1}^{K} \mathcal{I}_k) \cap Q_v| = 0$ for all $v \in \mathcal{O} \cap \mathcal{V}_R$, since $\|\hat{A}_v - \hat{A}_w\|^2 \geq \frac{1}{2}\|\hat{A}_v - M_\Gamma^k\|^2 - \|\hat{A}_w - M_\Gamma^k\|^2 > \frac{n\bar{p}^2}{\log(n\bar{p})}$ for all $w \in \mathcal{I}_k$;

(ii) $|\Gamma \setminus (\cup_{k=1}^{K} \mathcal{I}_k)| \leq \frac{\|\hat{A} - M_\Gamma\|_F^2}{\min_{v \in \Gamma \setminus (\cup_{k=1}^{K} I_k)} \|\hat{A}_v - M_\Gamma^k\|^2} = O\left(\frac{\log(n\bar{p})^3}{\bar{p}}\right)$;

(iii) $\mathcal{I}_k \subset Q_v$ for all $v \in \mathcal{I}_k \cap \mathcal{V}_R$, since $\|\hat{A}_v - \hat{A}_w\|^2 \leq 2\|\hat{A}_v - M_\Gamma^k\|^2 + 2\|\hat{A}_w - M_\Gamma^k\|^2 \leq \frac{n\bar{p}^2}{\log(n\bar{p})}$ for all $w \in \mathcal{I}_k$;

(iv) If $|Q_v \cap \mathcal{I}_k| \geq 1$, $|Q_v \cap \mathcal{I}_j| = 0$ for all $j \neq k$, since $\|M_\Gamma^k - M_\Gamma^j\| = \Omega(n\bar{p}^2)$ is much larger than the radius $\frac{n\bar{p}^2}{\log(n\bar{p})} = O(\frac{n\bar{p}^2}{\log(n\bar{p})})$;

From the properties of $\mathcal{I}_k$ and $\mathcal{O}$, we state the following results.

- From (i) and (ii), we deduce that

$$|Q_v| = O\left(\frac{\log(n\bar{p})^3}{\bar{p}}\right) \quad \text{for all} \quad v \in \mathcal{O} \cap V_R, \tag{29}$$

  since every $w \in (\cup_{k=1}^{K}\mathcal{I}_k)$ is outside of $Q_v$ (i.e., $w \in \Gamma \setminus (\cup_{k=1}^{K} I_k)$ is necessary for $w \in Q_v$);

- since $\alpha_k$ is a constant for all $k$ and $\frac{|\Gamma \setminus (\cup_{k=1}^{K}\mathcal{I}_k)|}{|\Gamma|} = o(1)$ from (ii), with high probability,

$$|\mathcal{I}_k \cap \mathcal{V}_R| \geq 1 \quad \text{for all} \quad 1 \leq k \leq K; \tag{30}$$

- The properties (ii), (iii), and (iv) and (30) imply that

$$|Q_v \setminus \cup_{l=0}^{k-1} S_l| \geq m_k, \quad \exists v \in (\cup_{m=1}^{K}\mathcal{I}_k \cap V_R) \setminus (\cup_{l=0}^{k-1} S_l), \tag{31}$$

  where $m_k$ is the $k$-th largest value among $\{|\mathcal{I}_1|, \ldots, |\mathcal{I}_K|\}$;

- since $|\mathcal{I}_k| \geq |V_k \cap (\Gamma \setminus \mathcal{O})| \geq \alpha_k n(1 - o(1))$ from (ii) and (iii),

$$|\mathcal{I}_k| \geq |V_k \cap (\Gamma \setminus \mathcal{O})| \geq \alpha_k n(1 - o(1)). \tag{32}$$

Thus, we can conclude that $\hat{K} = K$ from (31) and (32) and the property (ii); and from (29), there exists a permutation $\gamma$ such that $\|\hat{A}_{v_k^\star} - M_\Gamma^{\gamma(k)}\|^2 \leq 4\frac{n\bar{p}^2}{\log(n\bar{p})}$ for all $k$. Hence from (23), $\|\hat{A}_v - M_{v,\Gamma}\|^2 = \Omega\left(n\bar{p}^2\right)$ when $v$ is misclassified. ∎

## C.3   Proof of Theorem 2

From Chernoff bound, with high probability,

$$||\mathcal{V}_k| - \alpha_k n| \leq \sqrt{n}\log(n) \quad \text{for all} \quad k. \tag{33}$$

In what follows, we hence just prove the theorem assuming that (33) holds.

Let $H$ be the largest set of items $v \in \mathcal{V}$ satisfying:

(H1)  $e(v, \mathcal{V}) \leq 10\eta n\bar{p}L$,

(H2)  When $v \in \mathcal{V}_k$, $\sum_{i=1}^{K} \sum_{\ell=0}^{L} e(v, \mathcal{V}_i, \ell) \log \frac{p(k,i,\ell)}{p(j,i,\ell)} \geq \frac{n\bar{p}}{\log(n\bar{p})^4}$ for all $j \neq k$.

(H3)  $e(v, \mathcal{V} \setminus H) \leq 2\log(n\bar{p})^2$.

(H1) regularizes degrees, (H2) means that $v \in H$ is correctly classified when using the log-likelihood estimate, and (H3) means that $v$ does not share too many labels with items outside $H$.

The proof of the theorem follows from the following propositions. The first provides an upper bound of $|\mathcal{V} \setminus H|$, and the second provides the rate at which our estimated clusters improve in each iteration when we restrict our attention to items in $H$.

**Proposition 14** *When $nD(\alpha, p) - \frac{n\bar{p}}{\log(n\bar{p})^3} \geq \log(n/s) + \sqrt{\log(n/s)}$, $|\mathcal{V} \setminus H| \leq s$ with high probability.*

**Proposition 15** *If* $|\bigcup_{k=1}^{K}(S_k^{(0)} \setminus \mathcal{V}_k) \cap H| + |\mathcal{V} \setminus H| = O(1/\bar{p})$, *with high probability, the following statement holds*

$$\frac{|\bigcup_{k=1}^{K}(S_k^{(t+1)} \setminus \mathcal{V}_k) \cap H|}{|\bigcup_{k=1}^{K}(S_k^{(t)} \setminus \mathcal{V}_k) \cap H|} \leq \frac{1}{\sqrt{n\bar{p}}} \quad \text{for all} \quad t \geq 0.$$

From Proposition 15, after $\log(n)$ iterations (remember that $n\bar{p} = \omega(1)$, so when $n$ is large enough $1/\sqrt{n\bar{p}} \leq e^{-2}$), no item in $H$ can be misclassified with high probability. Hence the number of misclassified items cannot exceed $|\mathcal{V} \setminus H| \leq s$, $nD(\alpha, p) - \frac{n\bar{p}}{\log(n\bar{p})^3} \geq \log(n/s) + \sqrt{\log(n/s)}$. The proof is completed by remarking that if the previous condition on $D(\alpha, p)$ holds, then

$$1 \leq \lim_{n \to \infty} \frac{nD(\alpha, p) - \frac{n\bar{p}}{\log(n\bar{p})^3}}{\log(n/s) + \sqrt{\log(n/s)}} = \lim_{n \to \infty} \frac{nD(\alpha, p)}{\log(n/s)},$$

where we used $D(\alpha, p) = \Omega(\bar{p})$ from condition (A2) and Lemma 9. $\blacksquare$

### C.3.1 Proof of Proposition 14 – Size of $\mathcal{V} \setminus H$

We compute the number of items satisfying (H1), (H2), and (H3) in (34), (35), and Lemma 16, respectively.

Number of items satisfying (H1): From Lemma 11, we get:

$$\mathbb{P}\{e(v, \mathcal{V}) \leq 10\eta n\bar{p}L\} \geq 1 - \exp(-10\eta n\bar{p}L). \tag{34}$$

Number of items satisfying (H2): We shall prove that when $v$ satisfies (H1), $v$ satisfies (H2) as well with probability at least

$$1 - \exp\left(-nD(\alpha, p) + \frac{n\bar{p}}{2\log(n\bar{p})^3}\right). \tag{35}$$

To this aim, we first establish that if $v$ satisfies

$$\sum_{i=1}^{K} |\mathcal{V}_i| KL(\mu(v, \mathcal{V}_i), p(k, i)) \leq \left(1 - \frac{\log(n)^2}{\sqrt{n}}\right) nD(\alpha, p) - \frac{n\bar{p}}{\log(n\bar{p})^4}, \tag{36}$$

then $v$ satisfies (H2). Indeed, assume that (36) holds, then

(i) $\sum_{i=1}^{K} \alpha_i n KL(\mu(v, \mathcal{V}_i), p(k, i)) \leq \left(1 + \frac{\log(n)^2}{\sqrt{n}}\right) \sum_{i=1}^{K} |\mathcal{V}_i| KL(\mu(v, \mathcal{V}_i), p(k, i)) < nD(\alpha, p)$, since $||\mathcal{V}_i| - \alpha_i n| \leq \sqrt{n}\log(n)$ and (36) holds;

(ii) $\sum_{i=1}^{K} \alpha_i n KL(\mu(v, \mathcal{V}_i), p(j, i)) \geq nD(\alpha, p)$, since
$\max\left\{\sum_{i=1}^{K} \alpha_i KL(\mu(v, \mathcal{V}_i), p(j, i)), \sum_{i=1}^{K} \alpha_i KL(\mu(v, \mathcal{V}_i), p(k, i))\right\} \geq D(\alpha, p)$ and
$\sum_{i=1}^{K} \alpha_i KL(\mu(v, \mathcal{V}_i), p(k, i)) < D(\alpha, p)$;

(iii) $\sum_{i=1}^{K} |\mathcal{V}_i| KL(\mu(v, \mathcal{V}_i), p(j, i)) \geq \left(1 - \frac{\log(n)^2}{\sqrt{n}}\right) nD(\alpha, p)$, from ii) and the fact that $||\mathcal{V}_i| - \alpha_i n| \leq \sqrt{n}\log(n)$;

(iv) from (36) and iii), for all $j \neq i$,

$$\sum_{i=1}^{K} \sum_{\ell=0}^{L} e(v, \mathcal{V}_i, \ell) \log \frac{p(k, i, \ell)}{p(j, i, \ell)} = \sum_{i=1}^{K} |\mathcal{V}_i| \left(KL(\mu(v, \mathcal{V}_i), p(j, i)) - KL(\mu(v, \mathcal{V}_i), p(k, i))\right)$$
$$\geq \frac{n\bar{p}}{\log(n\bar{p})^4}.$$

Hence $v$ satisfies (H2). It remains to evaluate the probability of the event (36), which is done by applying Lemma 13 and proves (35).

Number of items satisfying (H3): From (34), (35), and the Markov inequality, we deduce that with probability at least $1 - \exp\left(-\sqrt{\log(n/s)}\right)$, the number of items that do not satisfy either (H1) or (H2) is less than $s/3$ when $nD(\alpha, p) - \frac{n\bar{p}}{\log(n\bar{p})^3} \geq \log(n/s) + \sqrt{\log(n/s)}$, since

$$\frac{\mathbb{E}\{\text{The number of items that do not satisfy either (H1) or (H2)}\}}{s/3}$$

$$\leq \frac{n\exp(-10\eta n\bar{p}L) + n\exp\left(-nD(\alpha, p) + \frac{n\bar{p}}{2\log(n\bar{p})^3}\right)}{s/3}$$

$$\leq \frac{n}{s}\exp\left(-nD(\alpha, p) + \frac{n\bar{p}}{\log(n\bar{p})^3}\right) \leq \exp\left(-\sqrt{\log(n/s)}\right), \tag{37}$$

where we have used Lemma 10 for the last inequality. Lemma 16 allows us to complete the proof of Proposition. $\blacksquare$

**Lemma 16** *When the number of items that do not satisfy either (H1) or (H2) is less than $s/3$, $|\mathcal{V} \setminus H| \leq s$, with high probability.*

*Proof.* Let $e(S, S) = \sum_{v \in S} e(S, S)$. Next we prove the following intermediate claim: there is no subset $S \subset \mathcal{V}$ such that $e(S, S) \geq s\log(n\bar{p})^2$ and $|S| = s$ with high probability. For any subset $S \in \mathcal{V}$ such that $|S| = s$, by Markov inequality,

$$\begin{aligned}
\mathbb{P}\{e(S, S) \geq s\log(n\bar{p})^2\} &\leq \inf_{t \geq 0} \frac{\mathbb{E}[\exp(e(S, S)t)]}{st\log(n\bar{p})^2} \\
&\leq \inf_{t \geq 0} \frac{\prod_{i=1}^{s^2/2}(1 + L\bar{p}\exp(t))}{st\log(n\bar{p})^2} \\
&\leq \inf_{t \geq 0} \exp\left(\frac{s^2 L\bar{p}}{2}\exp(t) - st\log(n\bar{p})^2\right) \\
&\leq \exp\left(-n\bar{p}s\left(\log n\bar{p} - \frac{sL}{2n}\exp\left(\frac{n\bar{p}}{\log n\bar{p}}\right)\right)\right) \\
&\leq \exp\left(-\frac{n\bar{p}s\log n\bar{p}}{2}\right), \tag{38}
\end{aligned}$$

where, in the last two inequalities, we have set $t = \frac{n\bar{p}}{\log n\bar{p}}$ and used the fact that: $\frac{n}{s} \geq \exp\left(\frac{n\bar{p}}{\log n\bar{p}}\right)$, which comes from the assumptions made in the theorem. Since the number of subsets $S \subset \mathcal{V}$ with size $s$ is $\binom{n}{s} \leq \left(\frac{en}{s}\right)^s$, from (38), we deduce:

$$\begin{aligned}
\mathbb{E}[|\{S : e(S, S) \geq s\log(n\bar{p})^2 \text{ and } |S| = s\}|] &\leq \left(\frac{en}{s}\right)^s \exp\left(-\frac{n\bar{p}s\log n\bar{p}}{2}\right) \\
&= \exp\left(-s\left(\frac{n\bar{p}\log n\bar{p}}{2} - \log\frac{en}{s}\right)\right) \\
&\leq \exp\left(-\frac{n\bar{p}s\log n\bar{p}}{4}\right).
\end{aligned}$$

Therefore, by Markov inequality, we can conclude that there is no $S \subset \mathcal{V}$ such that $e(S, S) \geq s\log(n\bar{p})^2$ and $|S| = s$ with high probability.

To conclude the proof of the lemma, we build the following sequence of sets. Let $Z_1$ denote the set of items that do not satisfy at least one of (H1) and (H2). Let $\{Z(t) \subset \mathcal{V}\}_{1 \leq t \leq t^\star}$ be generated as follows:

- $Z(0) = Z_1$.
- For $t \geq 1$, $Z(t) = Z(t-1) \cup \{v_t\}$ if there exists $v_t \in \mathcal{V}$ such that $e(v_t, Z(t-1)) > 2\log(n\bar{p})^2$ and $v_t \notin Z(t-1)$. If such an item does not exist, the sequence ends.

The sequence ends after the construction of $Z(t^\star)$. We show that if we assume that the cardinality of items that do not satisfy (H3) is strictly larger than $s/2$, then one the set of the sequence $\{Z(t) \subset \mathcal{V}\}_{1 \le t \le t^\star}$ contradicts the claim we just proved.

Assume that the number of items do not satisfy (H3) is strictly larger than $s/2$, then these items will be at some point added to the sets $Z(t)$, and by definition, each of these node contributes with more than $2\log(n\bar{p})^2$ in $e(Z(t), Z(t))$. Hence if starting from $Z_1$, we add $s/2$ items not satisfying (H3), we get a set $Z(t)$ of cardinality less than $s/3 + s/2$ and such that $e(Z(t), Z(t)) > s\log(n\bar{p})^2$. We can further add arbitrary items to $Z(t)$ so that it becomes of cardinality $s$, and the obtained set contradicts the claim. ∎

### C.3.2 Proof of Proposition 15

Recall that $\{S_j^{(t)}\}_{1 \le j \le K}$ is the partition after the $t$-th improvement iteration. Also recall that with loss of generality, we assume that the set of misclassified items in $H$ after the $t$-th step is $\mathcal{E}^{(t)} = \left(\cup_k (S_k^{(t)} \setminus \mathcal{V}_k)\right) \cap H$ (it should be defined through an appropriate permutation $\gamma$ of $\{1, \dots, K\}$ by $\mathcal{E}^{(t)} = (\cup_k (S_k^{(t)} \setminus \mathcal{V}_{\gamma(k)})) \cap H$, but we omit $\gamma$). With this notational convention, we can define $\mathcal{E}_{jk}^{(t)} = (S_j^{(t)} \cap \mathcal{V}_k) \cap H$ and $\mathcal{E}^{(t)} = \bigcup_{j,k:j \ne k} \mathcal{E}_{jk}^{(t)}$. At each improvement step, items move to the most likely cluster (according to the log-likelihood defined in the SP algorithm). Thus, for all $i$,

$$0 \le \sum_{j,k:j\ne k} \sum_{v\in\mathcal{E}_{jk}^{(t+1)}} \sum_{i=1}^{K}\sum_{\ell=0}^{L} e(v, S_i^{(t)}, \ell) \log \frac{\hat{p}(j,i,\ell)}{\hat{p}(k,i,\ell)}$$

$$\le \sum_{j,k:j\ne k} \sum_{v\in\mathcal{E}_{jk}^{(t+1)}} \sum_{i=1}^{K}\sum_{\ell=0}^{L} e(v, S_i^{(t)}, \ell) \log \frac{p(j,i,\ell)}{p(k,i,\ell)} + |\mathcal{E}^{(t+1)}|(n\bar{p})^{1-\kappa}\log(n\bar{p})^3 \qquad (39)$$

$$\le \sum_{j,k:j\ne k} \sum_{v\in\mathcal{E}_{jk}^{(t+1)}} \sum_{i=1}^{K}\sum_{\ell=0}^{L} e(v, \mathcal{V}_i, \ell) \log \frac{p(j,i,\ell)}{p(k,i,\ell)}$$

$$+ \sum_{w\in\mathcal{E}^{(t+1)}} e(w, \mathcal{E}^{(t)}) \log(2\eta) + 2|\mathcal{E}^{(t+1)}|(n\bar{p})^{1-\kappa}\log(n\bar{p})^3 \qquad (40)$$

$$\le -\frac{n\bar{p}}{\log(n\bar{p})^4}|\mathcal{E}^{(t+1)}| + \sum_{w\in\mathcal{E}^{(t+1)}} e(w, \mathcal{E}^{(t)}, \ell) \log(2\eta) + 2|\mathcal{E}^{(t+1)}|(n\bar{p})^{1-\kappa}\log(n\bar{p})^3 \quad (41)$$

$$\le -\frac{n\bar{p}}{\log(n\bar{p})^4}|\mathcal{E}^{(t+1)}| + \sqrt{|\mathcal{E}^{(t)}||\mathcal{E}^{(t+1)}|n\bar{p}\log n\bar{p}} + 3|\mathcal{E}^{(t+1)}|(n\bar{p})^{1-\kappa}\log(n\bar{p})^3. \qquad (42)$$

Therefore, from the above inequalities, we conclude that

$$\frac{|\mathcal{E}^{(t+1)}|}{|\mathcal{E}^{(t)}|} \le \frac{\log(n\bar{p})^{10}}{n\bar{p}} \le \frac{1}{\sqrt{n\bar{p}}}.$$

Next we prove all the steps of the previous analysis.

*Proof of* (39): From $\log(1+x) \le x$, when $p(j,i,\ell) - |\hat{p}(j,i,\ell) - p(j,i,\ell)| > 0$,

$$\left|\log \frac{\hat{p}(j,i,\ell)}{p(j,i,\ell)}\right| \le \frac{|\hat{p}(j,i,\ell) - p(j,i,\ell)|}{p(j,i,\ell) - |\hat{p}(j,i,\ell) - p(j,i,\ell)|}.$$

Thus, we just provide an upper bound of $|\hat{p}(j,i,\ell) - p(j,i,\ell)|$ to show (39). From the triangle inequality,

$$|\hat{p}(j,i,\ell) - p(j,i,\ell)|$$
$$= \frac{\left|e(S_i^{(0)}, S_j^{(0)}, \ell) - p(j,i,\ell)|S_i^{(0)}||S_j^{(0)}|\right|}{|S_i^{(0)}||S_j^{(0)}|}$$
$$\le \frac{\left|e(S_i^{(0)}, S_j^{(0)}, \ell) - \mathbb{E}[e(S_i^{(0)}, S_j^{(0)}, \ell)]\right| + \left|\mathbb{E}[e(S_i^{(0)}, S_j^{(0)}, \ell)] - p(j,i,\ell)|S_i^{(0)}||S_j^{(0)}|\right|}{|S_i^{(0)}||S_j^{(0)}|} \qquad (43)$$

We first find an upper bound of $\left| e(S_i^{(0)}, S_j^{(0)}, \ell) - \mathbb{E}[e(S_i^{(0)}, S_j^{(0)}, \ell)] \right|$. Let $\mathcal{S}$ be the of partitions such that

$$\left| \cup_{k=1}^K \mathcal{V}_k \setminus S_k \right| \leq \xi = O\left( \frac{\log(n\bar{p})^2}{\bar{p}} \right) \quad \text{for all} \quad \{S_k\}_{1 \leq k \leq K} \in \mathcal{S}.$$

Then,

$$
\begin{aligned}
|\mathcal{S}| &\leq \binom{n}{\xi} K^\xi \\
&\leq \left( \frac{ken}{\xi} \right)^\xi \\
&= \exp\left( O\left( \frac{\log(n\bar{p})^3}{\bar{p}} \right) \right).
\end{aligned}
\tag{44}
$$

For all $\{S_k\}_{1 \leq k \leq K} \in \mathcal{S}$ and for all $\ell \geq 1$ and $1 \leq i, j \leq K$, $e(S_i, S_j, \ell)$ is the sum of $|S_i||S_j|$ (or $\frac{|S_i|^2}{2}$ when $i = j$) independent Bernoulli random variables. Since the variance of $e(S_i, S_j, \ell)$ is always less than $n^2 \bar{p}$, by Chernoff inequality (e.g., Theorem 2.1.3 in [22]), with probability at least $1 - \exp\left( -\Theta\left( \frac{\log(n\bar{p})^4}{\bar{p}} \right) \right)$,

$$|e(S_i, S_j, \ell) - \mathbb{E}[e(S_i, S_j, \ell)]| \leq n\log(n\bar{p})^2 \quad \text{for all} \quad i, j, \ell. \tag{45}$$

From (44) and (45), with high probability,

$$|e(S_i, S_j, \ell) - \mathbb{E}[e(S_i, S_j, \ell)]| \leq n\log(n\bar{p})^2 \quad \text{for all} \quad i, j, \ell \quad \text{and} \quad \{S_k\}_{1 \leq k \leq K} \in \mathcal{S}.$$

Since $\{S_k^{(0)}\}_{1 \leq k \leq K} \in \mathcal{S}$, from the above inequality,

$$\left| e(S_i^{(0)}, S_j^{(0)}, \ell) - \mathbb{E}[e(S_i^{(0)}, S_j^{(0)}, \ell)] \right| \leq n\log(n\bar{p})^2 \quad \text{for all} \quad i, j, \ell. \tag{46}$$

We now devote to the remaining part of (43). Since $|\mathcal{E}^{(0)}| = O\left( \frac{\log(n\bar{p})^2}{\bar{p}} \right)$ from Theorem 7,

$$\left| \mathbb{E}[e(S_i^{(0)}, S_j^{(0)}, \ell)] - |S_i^{(0)}||S_j^{(0)}|p(i, j, \ell) \right| \leq \eta |\mathcal{E}^{(0)}| np(i, j, \ell) = O(n\log(n\bar{p})^2). \tag{47}$$

From (43), (46) and (47), with high probability,

$$|\hat{p}(j, i, \ell) - p(j, i, \ell)| = O(\log(n\bar{p})^2/n) \quad \text{for all} \quad i, j, \ell,$$

which implies that:

$$\left| \log \frac{\hat{p}(j, i, \ell)}{p(j, i, \ell)} \right| \leq \frac{|\hat{p}(j, i, \ell) - p(j, i, \ell)|}{p(j, i, \ell) - |\hat{p}(j, i, \ell) - p(j, i, \ell)|} = O\left( \frac{\log(n\bar{p})^2}{np(j, i, \ell)} \right) \quad \text{for all} \quad i, j, \ell.$$

Since $e(v, S_i^{(t)}, \ell) \leq e(v, \mathcal{V}) \leq 10\eta n\bar{p}L$ from (H1) and $np(j, i, \ell) \geq (n\bar{p})^\kappa$ from (A3), we deduce that, for all $v \in \Gamma$ and $i, j, k$,

$$\sum_{\ell=0}^L e(v, S_i^{(t)}, \ell) \left| \log \frac{\hat{p}(j, i, \ell)}{\hat{p}(k, i, \ell)} - \log \frac{p(j, i, \ell)}{p(k, i, \ell)} \right| = O\left( \log(n\bar{p})^2 (n\bar{p})^{1-\kappa} \right).$$

*Proof of* (40): Since $\log \frac{p(j,i,0)}{p(k,i,0)} = O(\bar{p})$ for all $i, j, k$ and $|\mathcal{E}^{(t)}| = O(\log(n\bar{p})^2/\bar{p})$,

$$\sum_{i=1}^{K}\sum_{\ell=0}^{L} e(v, S_i^{(t)}, \ell) \log \frac{p(j,i,\ell)}{p(k,i,\ell)}$$

$$= \sum_{i=1}^{K} \left( |S_i^{(t)}| \log \frac{p(j,i,0)}{p(k,i,0)} + \sum_{\ell=1}^{L} e(v, S_i^{(t)}, \ell) \log \frac{p(j,i,\ell)p(k,i,0)}{p(k,i,\ell)p(j,i,0)} \right)$$

$$\leq \sum_{i=1}^{K} \left( |\mathcal{V}_i| \log \frac{p(j,i,0)}{p(k,i,0)} + \sum_{\ell=1}^{L} e(v, S_i^{(t)}, \ell) \log \frac{p(j,i,\ell)p(k,i,0)}{p(k,i,\ell)p(j,i,0)} \right) + \log(n\bar{p})^3$$

$$\leq \sum_{i=1}^{K}\sum_{\ell=0}^{L} e(v, \mathcal{V}_i, \ell) \log \frac{p(j,i,\ell)}{p(k,i,\ell)} + \sum_{i=1}^{K}\sum_{\ell=1}^{L} e(v, \mathcal{V}_i \setminus S_i^{(t)}, \ell) \log(2\eta) + \log(n\bar{p})^3$$

$$= \sum_{i=1}^{K}\sum_{\ell=0}^{L} e(v, \mathcal{V}_i, \ell) \log \frac{p(j,i,\ell)}{p(k,i,\ell)} + \left( e(v, \mathcal{E}^{(t)}) + e(v, \mathcal{V} \setminus H) \right) \log(2\eta) + \log(n\bar{p})^3$$

$$\leq \sum_{i=1}^{K}\sum_{\ell=0}^{L} e(v, \mathcal{V}_i, \ell) \log \frac{p(j,i,\ell)}{p(k,i,\ell)} + \log(2\eta) e(v, \mathcal{E}^{(t)}) + 2\log(n\bar{p})^3,$$

where the last inequality stems from (H3), i.e., from $e(v, \mathcal{V} \setminus H) \leq 2\log(n\bar{p})^2$ when $v \in H$.

*Proof of* (41): Since $\mathcal{E}^{(t+1)} \subset H$ and every $v \in H$ satisfies (H2), every $v \in \mathcal{E}_{jk}^{(i+1)}$ satisfies:

$$\sum_{i=1}^{K}\sum_{\ell=0}^{L} e(v, \mathcal{V}_i, \ell) \log \frac{p(j,i,\ell)}{p(k,i,\ell)} \leq -\frac{n\bar{p}}{\log(n\bar{p})^4}.$$

*Proof of* (42): Let $\bar{\Gamma} = \{v : e(v, \mathcal{V}) \leq 10\eta n\bar{p}L\}$ and $A_{\bar{\Gamma}}^{\ell}$ be the trimmed matrix of $A^{\ell}$ whose elements in rows and columns corresponding to $w \notin \bar{\Gamma}$ are set to 0. $\bar{\Gamma}$ is the set of all items that satisfy (H1) and $H \subset \bar{\Gamma}$. Let $X_{\bar{\Gamma}} = \sum_{\ell=1}^{L}(A_{\bar{\Gamma}}^{\ell} - M_{\bar{\Gamma}}^{\ell})$. We have:

$$\sum_{v \in \mathcal{E}^{(t+1)}} \left( e(v, \mathcal{E}^{(t)}) - \mathbb{E}[e(v, \mathcal{E}^{(t)})] \right) \leq 1_{\mathcal{E}^{(t)}}^T \cdot X_{\bar{\Gamma}} \cdot 1_{\mathcal{E}^{(t+1)}},$$

where $1_S$ is the vector whose $v$-th component is equal to 1 if $v \in S$ and to 0 otherwise. Since $\mathbb{E}[e(v, \mathcal{E}^{(t)})] \leq \bar{p}L|\mathcal{E}^{(t)}|$ and $\|X_{\bar{\Gamma}}\|_2 \leq \sqrt{n\bar{p}\log n\bar{p}}$ with high probability from Lemma 12,

$$\sum_{v \in \mathcal{E}^{(t+1)}} e(v, \mathcal{E}^{(t)}) = \sum_{v \in \mathcal{E}^{(t+1)}} \left( e(v, \mathcal{E}^{(t)}) - \mathbb{E}[e(v, \mathcal{E}^{(t)})] \right) + \bar{p}L|\mathcal{E}^{(t)}||\mathcal{E}^{(t+1)}|$$

$$\leq \|1_{\mathcal{E}^{(t)}}^T \cdot X_{\bar{\Gamma}} \cdot 1_{\mathcal{E}^{(t+1)}}\|_2 + |\mathcal{E}^{(t+1)}|\log(n\bar{p})$$

$$\leq \|1_{\mathcal{E}^{(t)}}^T\|_2 \|X_{\bar{\Gamma}}\|_2 \|1_{\mathcal{E}^{(t+1)}}\|_2 + |\mathcal{E}^{(t+1)}|\log(n\bar{p})$$

$$\leq \sqrt{|\mathcal{E}^{(t)}||\mathcal{E}^{(t+1)}|n\bar{p}\log(n\bar{p})} + |\mathcal{E}^{(t+1)}|\log(n\bar{p}).$$

$\blacksquare$

# D  Proof of Theorem 3

The positive result is obtained by applying Theorem 2 to $s = \frac{1}{2}$. When $\liminf_{n \to \infty} \frac{nD(\alpha,p)}{\log(n)} \geq 1$, SP algorithm find clusters exactly with high probability. Thus, it suffices to show the negative result.

We prove the negative part by contradiction. Consider a maximum a posteriori (MAP) estimation with full parameter information. When we observe a labeld information $A$, the MAP estimates the clusters as follows:

$$(\hat{S}_k)_{k=1,...,k} = \arg \max_{(S_k)_{k=1,..,K}} \mathbb{P}\left\{ (S_k)_{k=1,..,K} | \alpha, p, K, A \right\}. \tag{48}$$

Let $\varepsilon^{\text{MAP}}$ denote the number of misclassified nodes by the MAP estimation. From the definition of the MAP estimation, for any clustring algorithm $\pi$, we have

$$\mathbb{P}\left\{\varepsilon^{\pi} \geq 1\right\} \geq \mathbb{P}\left\{\varepsilon^{\text{MAP}} \geq 1\right\}. \tag{49}$$

Thus, in what follows, we show that when $\liminf_{n\to\infty} \frac{nD(\alpha,p)}{\log(n)} < 1$, the MAP estimation is failed to find the exact clusters with high probability.

We start by Lemma 17 which finds a large deviation inequality for edge connections.

**Lemma 17** *Let $\boldsymbol{x} \in \mathbb{Z}^{K\times(L+1)}$ whose $(k, \ell+1)$ element is $x_{k,\ell}$, and such that $\sum_{\ell=0}^{L} x_{k,\ell} = |\mathcal{V}_k|$ for all $1 \leq k \leq K$, $\sum_{\ell=1}^{L} x_{k,\ell} = \Theta(n\bar{p})$ for all $k$, and*

$$\sum_{k=1}^{K} |\mathcal{V}_k| KL(\mu(v, \mathcal{V}_k), p(i,k)) = nD \quad when \quad e(v) = \boldsymbol{x},$$

*where we denote by $e(v)$ the $K \times (L+1)$ matrix whose $(k, \ell+1)$ element is $e(v, \mathcal{V}_k, \ell)$. Then,*

$$\log\left(\mathbb{P}\left\{e(v) = \boldsymbol{x}\right\}\right) \geq -nD(1 + o(1)) \quad when \quad v \in \mathcal{V}_i \quad and \quad D = \Omega(\bar{p}).$$

*Proof.* When using the convention $\sum_{\ell=a}^{b}$ as 0 when $a > b$, we have

$$\log\left(\mathbb{P}\left\{e(v) = \boldsymbol{x}\right\}\right)$$

$$= \sum_{k=1}^{K}\left(\left(|\mathcal{V}_k| - \sum_{\ell=1}^{L} x_{k,\ell}\right)\log\left(p(i,k,0)\right) + \sum_{\ell=1}^{L}\log\left(p(i,k,\ell)^{x_{k,\ell}}\binom{|\mathcal{V}_k| - \sum_{m=1}^{\ell-1} x_{k,m}}{x_{k,\ell}}\right)\right)$$

$$\geq \sum_{k=1}^{K}\left(\left(|\mathcal{V}_k| - \sum_{\ell=1}^{L} x_{k,\ell}\right)\log\left(p(i,k,0)\right) + \sum_{\ell=1}^{L}\log\left(p(i,k,\ell)^{x_{k,\ell}}\frac{\left(|\mathcal{V}_k| - \sum_{m=1}^{L} x_{k,m}\right)^{x_{k,\ell}}}{x_{k,\ell}!}\right)\right)$$

$$\overset{(a)}{\geq} \sum_{k=1}^{K}\left(\left(|\mathcal{V}_k| - \sum_{\ell=1}^{L} x_{k,\ell}\right)\log\left(p(i,k,0)\right) + \sum_{\ell=1}^{L}\log\left(\left(\frac{p(i,k,\ell)e}{\frac{x_{k,\ell}}{|\mathcal{V}_k| - \sum_{m=1}^{L} x_{k,m}}}\right)^{x_{k,\ell}}\frac{1}{e\sqrt{x_{k,\ell}}}\right)\right)$$

$$\overset{(b)}{=} \sum_{k=1}^{K}\left(\left(|\mathcal{V}_k| - \sum_{\ell=1}^{L} x_{k,\ell}\right)\log\left(p(i,k,0)\right) + \sum_{\ell=1}^{L}\log\left(\frac{p(i,k,\ell)e}{\frac{x_{k,\ell}}{|\mathcal{V}_k| - \sum_{m=1}^{L} x_{k,m}}}\right)^{x_{k,\ell}}\right) - o\left(\sum_{k=1}^{K}\sum_{\ell=1}^{L} x_{k,\ell}\right)$$

$$\overset{(c)}{\geq} \sum_{k=1}^{K}\left(|\mathcal{V}_k| - \sum_{\ell=1}^{L} x_{k,\ell}\right)\log\left(p(i,k,0)\left(1 + \frac{\sum_{\ell=1}^{L} x_{k,\ell}}{|\mathcal{V}_k| - \sum_{\ell=1}^{L} x_{k,\ell}}\right)\right)$$

$$+ \sum_{k=1}^{K}\left(\sum_{\ell=1}^{L} x_{k,\ell}\log\left(\frac{p(i,k,\ell)}{\frac{x_{k,\ell}}{|\mathcal{V}_k| - \sum_{m=1}^{L} x_{k,m}}}\right)\right) - o\left(\sum_{k=1}^{K}\sum_{\ell=1}^{L} x_{k,\ell}\right)$$

$$= \sum_{k=1}^{K}\left(|\mathcal{V}_k| - \sum_{\ell=1}^{L} x_{k,\ell}\right)\log\left(\frac{p(i,k,0)}{(|\mathcal{V}_k| - \sum_{\ell=1}^{L} x_{k,\ell})/|\mathcal{V}_k|}\right) + \sum_{k=1}^{K}\left(\sum_{\ell=1}^{L} x_{k,\ell}\log\left(\frac{p(i,k,\ell)}{x_{k,\ell}/|\mathcal{V}_k|}\right)\right)$$

$$+ \sum_{k=1}^{K}\left(\sum_{\ell=1}^{L} x_{k,\ell}\log\left(\frac{|\mathcal{V}_k| - \sum_{m=1}^{L} x_{k,m}}{|\mathcal{V}_k|}\right)\right) - o\left(\sum_{k=1}^{K}\sum_{\ell=1}^{L} x_{k,\ell}\right)$$

$$\overset{(d)}{\geq} -nD - o\left(\sum_{k=1}^{K}\sum_{\ell=1}^{L} x_{k,\ell}\right)$$

$$\overset{(e)}{\geq} -nD(1 + o(1)),$$

where $(a)$ is obtained from $n! \leq e\sqrt{n}\left(\frac{n}{e}\right)^{n}$; $(b)$ stems from $\sum_{k=1}^{K}\sum_{\ell=1}^{L} x_{k,\ell} = \omega(1)$; to derive $(c)$, we use $e^{\sum_{\ell=1}^{L} x_{k,\ell}} \geq \left(1 + \frac{\sum_{\ell=1}^{L} x_{k,\ell}}{|\mathcal{V}_k| - \sum_{\ell=1}^{L} x_{k,\ell}}\right)^{|\mathcal{V}_k| - \sum_{\ell=1}^{L} x_{k,\ell}}$ since $e \geq (1 + 1/x)^{x}$ for all $x > 0$; to

prove $(d)$, we use the definition of $\boldsymbol{x}$ and the following inequality:

$$\sum_{\ell=1}^{L} x_{k,\ell} \log \left( \frac{|\mathcal{V}_k|}{|\mathcal{V}_k| - \sum_{m=1}^{L} x_{k,m}} \right) = \frac{\left( \sum_{\ell=1}^{L} x_{k,\ell} \right)^2}{|\mathcal{V}_k| - \sum_{\ell=1}^{L} x_{k,\ell}} (1 + o(1)) = o(\sum_{\ell=1}^{L} x_{k,\ell});$$

and $(e)$ is obtained from the definition of $\boldsymbol{x}$ that $\sum_{\ell=1}^{L} x_{k,\ell} = \Theta(n\bar{p})$ for all $k$. ∎

Assume that there exists a constant $\eta > 0$ such that $\frac{nD(\alpha,p)}{\log(n)} < 1 - \eta$.

Let $(i^\star, j^\star) = \arg\min_{i,j:i<j} D_{L+}(p(i), p(j))$ (i.e., it is the hardest case to discriminate cluster $i^\star$ and cluster $j^\star$). When $n \to \infty$, one can easily check using the continuity of the KL divergence that there exists $\boldsymbol{x}^\star$ such that when $e(v) = \boldsymbol{x}^\star$,

$$\frac{\eta}{2} \log n + \sum_{k=1}^{K} |\mathcal{V}_k| KL(\mu(v, \mathcal{V}_k), p(j^\star, k)) < \sum_{k=1}^{K} |\mathcal{V}_k| KL(\mu(v, \mathcal{V}_k), p(i^\star, k)) \quad \text{and} \quad (50)$$

$$\sum_{k=1}^{K} |\mathcal{V}_k| KL(\mu(v, \mathcal{V}_k), p(i^\star, k)) \leq (1 - \eta/2) \log(n). \quad (51)$$

Let $\mathcal{V}_e = \{v \in \mathcal{V}_{i^\star} : e(v) = \boldsymbol{x}^\star\}$. From (51) and Lemma 17, $\mathbb{E}[|\mathcal{V}_e|] \geq n^{\eta/4}$. Thus, from Markov inequality, with probability at least $1 - n^{-\eta/4}$, $\mathcal{V}_e$ is not empty (i.e., $|\mathcal{V}_e| \geq 1$).

Let $v^\star \in \mathcal{V}_e$ be a node in $\mathcal{V}_e$. We denote by $\Phi$ the original partition and define a slightly modified partition $\Psi$ as follows:

$$\hat{\mathcal{V}}_{i^\star} = \mathcal{V}_{i^\star} \setminus \{i^\star\}, \quad \hat{\mathcal{V}}_{j^\star} = \mathcal{V}_{j^\star} \cup \{i^\star\}, \quad \text{and} \quad \hat{\mathcal{V}}_k = \mathcal{V}_k \quad \text{otherwise.}$$

Then, $\Psi$ is a more likely partition than $\Phi$ from (50), i.e.,

$$\mathbb{P}\{\Phi|\alpha, p, K, A\} \geq \mathbb{P}\{\Psi|\alpha, p, K, A\} \quad (52)$$

which means that the MAP estimator does not select the exact partition when $\mathcal{V}_e$ is not empty. Therefore, from (49), every clustering algorithm $\pi$ has the error probability that

$$\mathbb{E}\{\varepsilon^\pi \geq 1\} \geq 1 - n^{-\eta/4}$$

when there exists a constant $\eta > 0$ such that $\frac{nD(\alpha,p)}{\log(n)} < 1 - \eta$. ∎

# E   Proof of Claim 6

When $\bar{p} = o(\frac{1}{\sqrt{n}})$, we have from Lemmas 11 and 13:

$$\lim_{n\to\infty} \frac{\log \left( \mathbb{P}\left\{ \sum_{k=1}^{K} |\mathcal{V}_k| KL\left( \mu(v, \mathcal{V}_k), p(i,k) \right) \geq nD \right\} \right)}{nD}$$

$$\leq \lim_{n\to\infty} \frac{\log \left( \mathbb{P}\left\{ \sum_{k=1}^{K} |\mathcal{V}_k| KL\left( \mu(v, \mathcal{V}_k), p(i,k) \right) \geq nD, e(v, \mathcal{V}) \leq 10\eta n\bar{p}L \right\} + \mathbb{P}\{e(v, \mathcal{V}) \geq 10\eta n\bar{p}L\} \right)}{nD}$$

$$\leq \lim_{n\to\infty} \frac{\log \left( \exp\left( -nD(1 - o(1)) \right) + \exp(-10\eta n\bar{p}L) \right)}{nD}$$

$$\leq -1.$$

Thus, to prove the result, it suffices to show that:

$$\lim_{n\to\infty} \frac{\log \left( \mathbb{P}\left\{ \sum_{k=1}^{K} |\mathcal{V}_k| KL\left( \mu(v, \mathcal{V}_k), p(i,k) \right) \geq nD \right\} \right)}{nD} \geq -1.$$

Next, we denote by $e(v)$ the $K \times (L + 1)$ matrix whose $(k, \ell + 1)$ element is $e(v, \mathcal{V}_k, \ell)$. Let $\boldsymbol{x} \in \mathbb{Z}^{K \times (L+1)}$ whose $(k, \ell + 1)$ element is $x_{k,\ell}$, and such that $\sum_{\ell=0}^{L} x_{k,\ell} = |\mathcal{V}_k|$ for all $1 \leq k \leq K$, $\sum_{\ell=1}^{L} x_{k,\ell} = \Theta(n\bar{p})$ for all $k$, and

$$\sum_{k=1}^{K} |\mathcal{V}_k| KL(\mu(v, \mathcal{V}_i), p(i,k)) = nD(1 + o(1)) \quad \text{when} \quad e(v) = \boldsymbol{x}.$$

We can easily check using the continuity of the KL divergence that such a choice for $x$ is possible. Then, from Lemma 17,

$$\log\left(\mathbb{P}\left\{\sum_{k=1}^{K}|\mathcal{V}_k|KL\left(\mu(v,\mathcal{V}_k),p(i,k)\right)\geq nD\right\}\right)\geq\log\left(\mathbb{P}\left\{e(v)=x\right\}\right)$$
$$\geq -nD(1+o(1)).$$

∎