[Reviews · NeurIPS 2016]

Reviewer 1

Summary

The paper considers the fairly general variant of the Stochastic Block model, called Labeled Stochastic Block Model, where the edges have one of \ell labels (the usual SBM corresponds to the case of two labels -- 0 and 1). The paper characterizes the optimal conditions involving the parameters p(i,j,\ell), K etc. so that we can recover all by s=o(n) of the vertices. This applies in the setting when the average-degree is \omega(1). The paper presents a lower bound, and an upper bound using an algorithm based on spectral algorithms+refinement. These two results together give the optimal results for the labeled block models and its special cases (like usual SBMs) for partial recovery.

Qualitative Assessment

The paper settles the problem of partial recovery (up to s=o(n) vertices) in the case of Labeled SBM, by presenting both a lower bound and a spectral algorithm. Getting sharp guarantees of this kind was not known even for SBMs. The tools and techniques are somewhat similar to the work of Abbe-Sandon on SBMs; however, it is technically impressive that they are able to get these sharp guarantees. The main drawback of these results is that the algorithms are tailor-made for the Labeled SBM model and do not seem robust to be practical. Comments: 1. It would be good to compare the results on Recovering a single hidden community in page 5 with old results on finding Planted Dense k-subgraphs [BCCFV10], which use very different algorithms in the regime when the degree and k are > n^{\Omega(1)}. 2. It would be good to distinguish the Labeled Stochastic Block model from Correlation Clustering [Bansal et al.,Mathieu-Schudy]/ Z_2 synchronization, where each edge has a +/- sign that has a small bias bounded away from 1/2.

Confidence in this Review

2-Confident (read it all; understood it all reasonably well)


Reviewer 2

Summary

A generalization of the well-studied stochastic block model (SBM) is the labeled SBM. In this model, two elements also have a label assigned to them denoting their similarity level. The known thresholding result by Abbe et al. (FOCS 2015) for SBM is generalized for the labeled case in this paper.

Qualitative Assessment

The paper come across to me as an extension of known results for the SBM. I do not see any unexpected result - or that has not been specifically pointed out in the introduction. Is their any theoretical innovation that had to be made for these extensions? Such as new tools from information theory, or statistics? Also if there is such, is it worth going through al the effort? Because the result seems like a straightforward extension to Abbe et al.'s result. Similar observation can be made for the generalized spectral algorithm, which is even claimed to be simple extension (line 70).

Confidence in this Review

2-Confident (read it all; understood it all reasonably well)


Reviewer 3

Summary

This paper explores the fundamental recovery limits of community recovery in a generalized version of the stochastic block model, where each community has size linear in n. The measurements between each pair of vertices can be one of L labels, which allows to model the SBM, the planted clique problem, and many others using a unified model. The focus is exact recovery or strong recovery (for which one expects the portion of misclassified nodes to be asymptotically zero). The theoretical results are quite strong, which generalize many of the prior works along this direction. The model is also of a lot of interest to multiple communities, so the contribution is timely and important. I think this is worth being accepted to NIPS. Despite the strong theory, I am not completely satisfied by the way this paper is written. There are many key assumptions / metrics that have not been given enough interpretation, so it is hard for non-experts to understand what these results / conditions really mean. The main algorithms are not described in the main text, and there are no simulations carried out (so it's hard to see how much practical relevance the theory bears). I hope that the authors can consider my comments when revising their paper.

Qualitative Assessment

1. The key conditions (A1) and (A2) need more interpretation. How critical are these assumptions? Are they coming from computational considerations or statistical considerations? Will the fundamental information limits change without these two assumptions? 2. The authors start by introducing "labels" in the SBMs but defer all motivations / applications to much later. I think the paper would read better if the authors at least describe some concrete applications right after (or before) describing the mathematical setup. This would be particularly beneficial for those who are non-experts in this field. 3. Some interpretation of the divergence metric D_{L+} is missing, as this is the most important metric that dictates the recovery limits. This metric is not easy to parse especially for those who have not read [2,3]. It might be nice to give some simple examples to interpret it. 4. The achievability part (i.e. Theorem 2) operates upon another assumption (A3), but I do think this additional assumption needs further interpretation as well. 5. I understand that Algorithms 1 and 2 are deferred to the supplemental materials due to space limitation, but I think these are very crucial for the paper and deserve a place in the main text (at least I think presenting the procedures is more helpful than elaborating the proof techniques in such a short conference paper). 6. There is no simulation carried out in this paper, so it is hard to see how well the derived information limits match practice (e.g. how large does n need to be in order for the practical phase transition to match the theoretical prediction).

Confidence in this Review

2-Confident (read it all; understood it all reasonably well)


Reviewer 4

Summary

The paper considers the problem of clustering in Labeled SBM for a finite number of clusters, all of size linear in n and provides several interesting results. For the case considered, for any given s = o(n), the paper provides both necessary and sufficient conditions for guaranteed clustering with at most s items that are in error. As a consequence, the paper also gives a necessary and sufficient conditions on the parameters of the model required for any algorithm to guarantee exact clustering under the regime of linearly growing clusters. With further assumptions it also shows that a certain spectral algorithm can with high probability recover the clusters exactly.

Qualitative Assessment

1) The paper considers the case when all the clusters are growing linearly in n (and hence K is finite). Is this a fundamental bottleneck or an artifact the proof technique? 2) Theorem 2 apart from (A3) also assumes p = o(1) (compared to Theorem 1). What happens if we tolerate p(i, j, l) that do not depend on n? 3) As a result of the assumption that all clusters are growing linearly in n, Theorem 3 for L = 2 gives suboptimal result for minimum cluster size (which is a bottleneck for clustering algorithms). In particular, the minimum cluster size has to be \Omega(n). The analysis of convex algorithms for graph clustering (Ames et. al 2010, Jalali et. al 2011, Vinayak et. al 2014,...) provides exact recovery guarantees for K = O(\sqrt{n}) while allowing heterogeneous cluster sizes i.e., there can be some clusters that grow linearly and others grow sublinearly (upto \Omega(\sqrt{n})) in n. Guarantees for Spectral clustering (Mc Sherry 2001, Rohe et. al 2011, Balakrishnan et al. 2011,...) allow K = O(\sqrt{n}) as long as the clusters are all within constant size of each other. In both the cases (convex algorithms and spectral clustering), p and q in SBM can be as small as Omega(polylog(n) / n). Minor point: It would be better to have the algorithm in the paper since a part of the paper is about guarantees for it. There is enough white space in the paper to be able to fit it in. In summary, the paper considers clustering with Labeled SBM in the regime of finite number of clusters, all of which grow linearly in n and provide necessary and sufficient conditions that guarantee clustering upto at most s = o(n) errors. The paper also relate the divergence defined for their results to other results in this regime. The paper is very well written.

Confidence in this Review

2-Confident (read it all; understood it all reasonably well)


Reviewer 5

Summary

This paper develops theoretical boundaries for detecting communities in labeled stochastic block model, which can be regarded as a generalization of the classic stochastic block model. Their result connects several work on recovery consistency conditions in block models. A variant of spectral partition algorithm is proposed, which achieves the bound before.

Qualitative Assessment

The authors analyze a family of models in a unified framework and give answer to the question when one could recover a consistent cluster assignment (weakly or strongly) from the observation. The result is of much importance but the presentation and organization could be improved. For example, as Theorem 2 refers to Algorithm 1, it is better to include Algorithm 1 in the main paper to keep it self-contained. Another suggestion is, since the authors compare themselves with many existing result, it's very helpful if all those comparisons can be formalized as corollaries and improvements are shown explicitly. Also please define all the notations used in the paper (such as \omega(1)), and avoid using the same letters for referring different quantities (like the two $w$ in line 236). I'm also interested in some simulations which compare the algorithm with existing ones since the authors mention about computational efficiency.

Confidence in this Review

2-Confident (read it all; understood it all reasonably well)


Reviewer 6

Summary

This paper makes contributions to the existing literature by extending the exact recovery(every single node is correctly recovered) results for the Stochastic Block Model in two ways. One contribution is they provide a proof for when all but a constant number of nodes can be recovered in the SBM. The second contribution is that they generalize the model to allow for multiple types of edges which causes them to rename the SBM as the Labeled SBM. They also provide an algorithm that achieves their proven recovery bounds.

Qualitative Assessment

Overall this paper generalizes the exact recovery results nicely. However, the results of this paper, like the other exact recovery results, are concerned with a constant number of errors in a graph where the total number of nodes is growing to infinity. The meaningful difference between 5/10/15 errors when the overall number and degree of the nodes is growing to infinity seems small to me. That being said this paper does seem to provide sound proofs for this question which is definitely not an easy question to answer. These proofs, and the outlines of insights behind these proofs, were not very clearly explained. Specifically the change of measure and introduction of Q were not very intuitively introduced. However, the algorithm portion was explained completely and clearly. Also, the Related Work review was not always accurate and did not contextualize this work as well as it could have. For example the reference [6] does not deal with constant average degree but also deals with the case when the average degree is going to infinity similar to this paper (they are different from this paper by requiring some noise measure lambda to be O(1)). I also think it would have been helpful for context to explain the similarities and differences between this proof and exact recovery proofs beyond the context that they provide in claim 4.

Confidence in this Review

2-Confident (read it all; understood it all reasonably well)